# Metabolic potential of uncultured bacteria and archaea associated with petroleum seepage in deep-sea sediments

Xiyang Dong [1], Chris Greening [2], Jayne E. Rattray[1], Anirban Chakraborty [1], Maria Chuvochina[2], Daisuke Mayumi[1,3], Jan Dolfing [4], Carmen Li[1], James M. Brooks[5], Bernie B. Bernard[5], Ryan A. Groves[1], Ian A. Lewis[1] & Casey R.J. Hubert[1]

The lack of microbial genomes and isolates from the deep seabed means that very little is known about the ecology of this vast habitat. Here, we investigate energy and carbon acquisition strategies of microbial communities from three deep seabed petroleum seeps (3 km water depth) in the Eastern Gulf of Mexico. Shotgun metagenomic analysis reveals that each sediment harbors diverse communities of chemoheterotrophs and chemolithotrophs. We recovered 82 metagenome-assembled genomes affiliated with 21 different archaeal and bacterial phyla. Multiple genomes encode enzymes for anaerobic oxidation of aliphatic and aromatic compounds, including those of candidate phyla Aerophobetes, Aminicenantes, TA06 and Bathyarchaeota. Microbial interactions are predicted to be driven by acetate and molecular hydrogen. These findings are supported by sediment geochemistry, metabolomics, and thermodynamic modelling. Overall, we infer that deep-sea sediments experiencing thermogenic hydrocarbon inputs harbor phylogenetically and functionally diverse communities potentially sustained through anaerobic hydrocarbon, acetate and hydrogen metabolism.

---

[1] Department of Biological Sciences, University of Calgary, Calgary, AB T2N 1N4, Canada. [2] School of Biological Sciences, Monash University, Clayton, VIC 3800, Australia. [3] Institute for Geo-Resources and Environment, Geological Survey of Japan, National Institute of Advanced Industrial Science and Technology (AIST), 1-1-1 Higashi, Tsukuba 305-8567, Japan. [4] School of Engineering, Newcastle University, Newcastle upon Tyne NE1 7RU, UK. [5] TDI Brooks International, College Station, TX 77845, USA. Correspondence and requests for materials should be addressed to X.D. (email: xiyang.dong@ucalgary.ca) or to C.R.J.H. (email: chubert@ucalgary.ca)

Deep-sea sediments, generally understood to be those occurring in water depths greater than ~500 m, represent one of the largest habitats on Earth. In recent years, culture-independent 16S rRNA gene surveys and metagenomic studies have revealed these sediments host a vast abundance and diversity of bacteria and archaea[1–6]. Cell numbers decrease with sediment depth and age, from between $10^6$ and $10^{10}$ cm$^{-3}$ in the upper cm at the sediment–water interface to below $10^4$ cm$^{-3}$ several kilometers below the ocean floor[7]. However, due to a lack of cultured representatives and genomes recovered from deep-sea sediments, it remains largely unresolved how microorganisms survive and function in these nutrient-limited ecosystems. Energy and carbon sources are essential requirements that allow the buried microorganisms to persist. With sunlight not reaching the deep seabed, photosynthetic processes do not directly support these communities[8]. It has therefore been proposed that deep-sea benthic and subseafloor microbes are primarily sustained by complex detrital organic matter, including carbohydrates, proteinaceous compounds, and humic substances, derived from the overlying water column via sedimentation[8,9].

Another important potential carbon and energy sources in deep-sea sediments are petroleum geofluids that migrate from subsurface reservoirs up to the seafloor. Petroleum compounds include smaller gaseous molecules, such as methane, propane, and butane, and larger aliphatic and aromatic liquids. Numerous studies have investigated the role of methane oxidation in seabed sediments, which is mediated by anaerobic methanotrophic archaea (ANME), generally in syntrophy with bacteria respiring sulfate or other electron acceptors[2,4,6,10]. In contrast, little is known about the degradation of larger alkanes or aromatic compounds by deep seabed microorganisms. Vigneron et al.[1] performed a comparative gene-centric study of hydrocarbon and methane seeps in the Gulf of Mexico, and suggested that microorganisms in deep cold seeps (water depth ~1 km) can potentially utilize a range of nonmethane hydrocarbons. However, due to the absence of metagenome binning in that study, relevant metabolic functions were not assigned to specific pathways or taxa.

In addition to organic carbon compounds, microbial life in deep-sea sediments is also supported by inorganic electron donors. Some microorganisms have been isolated from deep sediments that are able to sustain themselves by oxidizing elemental sulfur, hydrogen sulfide, carbon monoxide, ammonia, and molecular hydrogen ($H_2$)[4,6,8]. Of these, $H_2$ is a particularly important energy source given its production in large quantities by biological and geochemical processes. $H_2$ can be generated as a metabolic byproduct of fermentation, together with volatile fatty acids such as acetate, during organic matter degradation[7]. $H_2$ can also be produced abiotically via serpentinization, radiolysis of water, or thermal alteration of sedimentary organic matter[11]. For example, the radiolysis of water by naturally occurring radionuclides (e.g. $^{40}K$ and $^{238}U$) is estimated to produce $10^{11}$ mol $H_2$ per year[6,12]. Depending on the availability of electron acceptors, $H_2$ oxidation can be coupled to sulfate, nitrate, metal, and organohalide respiration, as well as acetogenesis and methanogenesis[6,8].

In this study, we used culture-independent approaches to study the role of microbial communities in the degradation of organic matter, including both detrital biomass and petroleum hydrocarbons. We performed metagenomic, geochemical and metabolomic analyses of deep seabed sediments (water depth ~3 km). Samples were chosen from three sites exhibiting evidence of different levels of migrated thermogenic hydrocarbons. Metagenomes generated from sediment samples of each site were assembled and binned to obtain metagenome-assembled genomes (MAGs) and to reconstruct metabolic pathways for dominant

members of the microbial communities. Complementing this genome-resolved metagenomics, a gene-centric analysis was performed by directly examining unassembled metagenomic data. Through the combination of metagenomics with geochemistry and metabolomics, supported by thermodynamic modeling, we provide evidence that (1) deep-sea sediments harbor phylogenetically diverse heterotrophic and lithotrophic microbial communities; (2) some members from the candidate phyla are engaged in degradation of aliphatic and aromatic compounds; and (3) microbial community members are likely interconnected via acetate and hydrogen metabolism.

## Results

**Hydrocarbon migration in seabed sediments.** This study tested three petroleum-associated near-surface sediments (referred to as Sites E26, E29, and E44; see map in Supplementary Fig. 1) sampled from the Eastern Gulf of Mexico[13]. Migrated thermogenic hydrocarbon content in the piston cores was analyzed for each of the three sites (Table 1). All three sediments contained high concentrations of aromatic compounds and liquid alkanes; aromatic compounds were most abundant at Site E26, while liquid alkanes were on average 2.5-fold higher concentration at Sites E26 and E29 than Site E44. Alkane gases were only abundant at Site E29 and were almost exclusively methane ($CH_4$). $CH_4$ sources can be inferred from stable isotopic compositions of $CH_4$ and molar ratios of $CH_4$ to higher hydrocarbons[10]. Ratios of $C_1$/($C_2 + C_3$) were greater than 1000 and $\delta^{13}C$ values of methane were more negative than −60‰, indicating that the $CH_4$ in these sediments was predominantly biogenic[10]. Similar co-occurrence of biogenic methane and complex hydrocarbons have been reported in a nearby seep in the Mississippi Canyon in the Gulf of Mexico[14]. GC-MS revealed an unresolved complex mixture (UCM) of saturated hydrocarbons in the $C_{15+}$ range in all three sites. Such UCM signals correspond to degraded petroleum hydrocarbons and may indicate the occurrence of oil biodegradation at these sites[15].

**Phylogenetically diverse bacterial and archaeal communities.** Illumina NextSeq sequencing of genomic DNA from deep-sea sediment communities produced 85,825,930, 148,908,270, and 138,795,692 quality-filtered reads for Sites E26, E29, and E44, respectively (Supplementary Table 1). The 16S rRNA gene amplicon sequencing results suggest that the sediments harbor diverse bacterial and archaeal communities, with Chao1 richness estimates of 359, 1375, and 360 amplicon sequence variants (ASVs) using bacterial-specific primers, and 195, 180 and 247 ASVs using archaeal-specific primers, for Sites E26, E29, and E44, respectively (Supplementary Table 2 and Supplementary Fig. 2). In accordance with amplicon sequencing results, taxonomic profiling of metagenomes using small subunit ribosomal RNA (SSU rRNA) marker genes demonstrated that the most abundant phyla in the metagenomes were, in decreasing order, Chloroflexi (mostly classes *Dehalococcoidia* and *Anaerolineae*), *Candidatus* Atribacteria, Proteobacteria (mostly class *Deltaproteobacteria*), and *Candidatus* Bathyarchaeota (Supplementary Data 1 and Fig. 1a). While the three sites share a broadly similar community composition, *Ca.* Bathyarchaeota and Proteobacteria were notably in higher relative abundance at the sites associated with more hydrocarbons (E29 and E26; Table 1), whereas the inverse is true for Actinobacteria, the Patescibacteria group, and *Ca.* Aerophobetes that are all present in higher relative abundance at Site E44 where associated hydrocarbon levels are lower. Additional sampling is required to determine whether these differences are due to the presence of hydrocarbons or other factors.

Assembly and binning for the three metagenomes resulted in a total of 82 MAGs with >50% completeness and <10% contamination based on CheckM analysis[16]. Reconstructed MAGs comprise taxonomically diverse members from a total of six archaeal and 15 bacterial phyla (Fig. 2 and Supplementary Data 2). Within the domain Bacteria, members of the phylum Chloroflexi are highly represented in each sample, especially from

the classes *Dehalococcoidia* and *Anaerolineae*. Within the domain Archaea, members of phylum Bathyarchaeota were recovered from all three sites. Most other MAGs belong to poorly understood candidate phyla that lack cultured representatives, including Aminicenantes (formerly OP8), Aerophobetes (formerly CD12), Cloacimonas (formerly WWE1), Stahlbacteria (formerly WOR-3), Atribacteria (formerly JS1 and OP9), TA06 (Supplementary Note 1 and Supplementary Data 3), and the Asgard superphylum, including Lokiarchaeota, Thorarchaeota, and Heimdallarchaeota.

In summary, while there are considerable community-level differences between the three sample locations, the recovered MAGs share common taxonomic affiliations at the phylum and class levels. Guided by associated geochemistry from the three sediment cores (Table 1 and Supplementary Note 2), we subsequently analyzed the metabolic potential of these MAGs to understand how bacterial and archaeal community members generate energy and biomass in these natural petroleum-associated deep-sea environments. Hidden Markov models (HMMs) and homology-based models were used to search for the presence of different metabolic genes in both the recovered MAGs and unbinned metagenomes. Where appropriate, findings were further validated through metabolomic analyses, phylogenetic visualization, and analysis of gene context.

### Detrital biomass and hydrocarbon degradation.

In deep-sea marine sediments, organic carbon is supplied either as detrital

**Table 1 Geochemical description of sediment samples from Sites E26, E29, and E44**

| Core ID | Site E26 | Site E29 | Site E44 |
|---|---|---|---|
| Latitude (N) | 26.59 | 27.43 | 26.28 |
| Longitude (W) | 87.51 | 86.01 | 86.81 |
| Water depth (km) | 2.8 | 3.2 | 3.0 |
| TSF Max | 57,326.7 | 26,738.3 | 13,502.3 |
| UCM ($\mu g\,g^{-1}$) | 32 | 13 | 7.3 |
| $\Sigma$n-Alkanes ($ng\,g^{-1}$) | 2845.3 | 2527 | 1045 |
| T/D ratio | 1.0 | 2.6 | 0.8 |
| $\Sigma$Alkane gas (ppm) | 9 | 36,012 | 9.9 |
| $C_{2+}$ Alkanes (ppm) | 0.3 | 17.5 | 0.5 |
| $C_1/(C_2 + C_3)$ | NA | 3974.2 | NA |
| $\delta^{13}CH_4$ (‰, vs. PDB)[a] | NA | −85.1 | NA |

*TSF Max* total scanning fluorescence maximum intensity. *UCM* unresolved complex mixture. *Σn-Alkanes* sum of $C_{15}$—$C_{34}$ n-alkanes. *ΣAlkane Gas* total alkane gases. *$C_{2+}$ Alkanes* sum of alkane gases larger than methane. *T/D* thermogenic/diagenetic n-alkane
[a] The $\delta^{13}CH_4$ values in Sites E26 and E29 could not be determined due to low methane concentration, which can be approximated by subtracting $C_{2+}$ Alkanes from ΣAlkane Gas

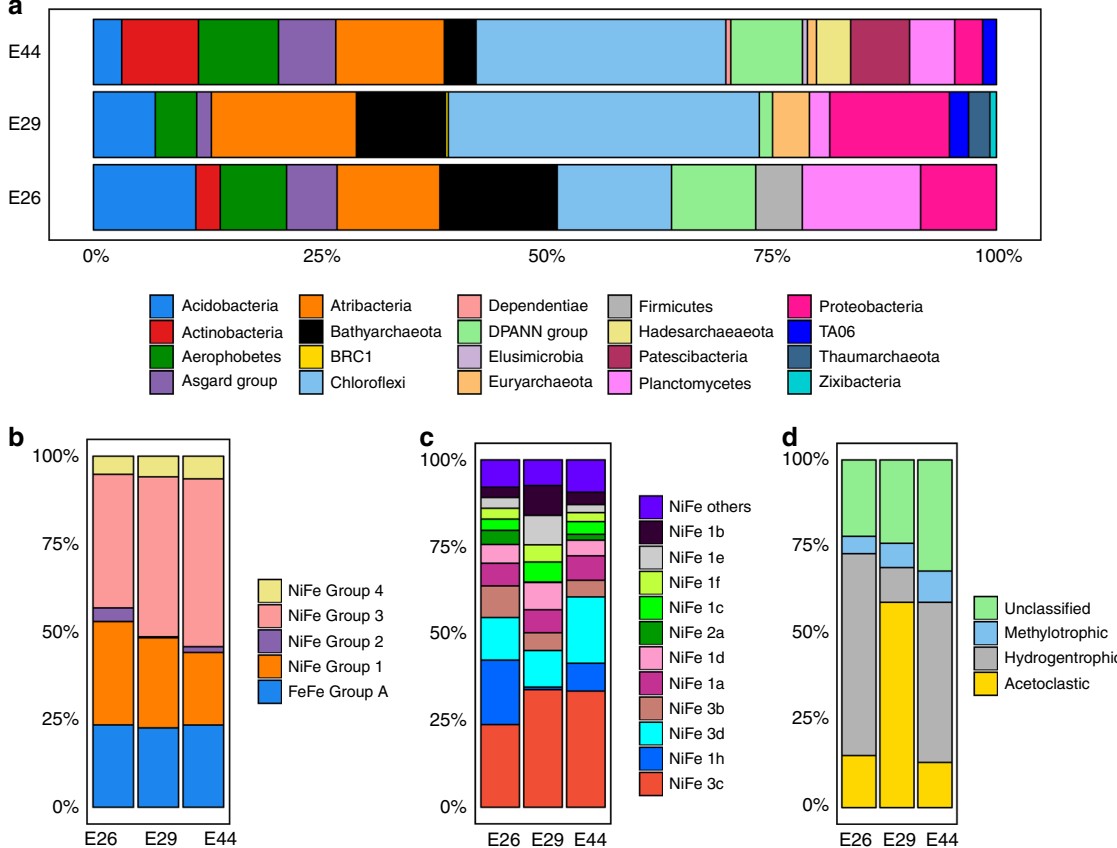

**Fig. 1** Relative frequency of metagenomic reads for different marker genes at Sites E26, E29, and E44. **a** Community composition based on reconstruction of full-length 16S rRNA genes from the metagenomes. Eukaryotes and unassigned reads are not shown. **b** Relative occurrences of hydrogenases with different metal cofactors. **c** Relative abundance of different subtypes of NiFe hydrogenases. **d** Relative abundance of *mcrA* genes indicative of different types of methanogenesis

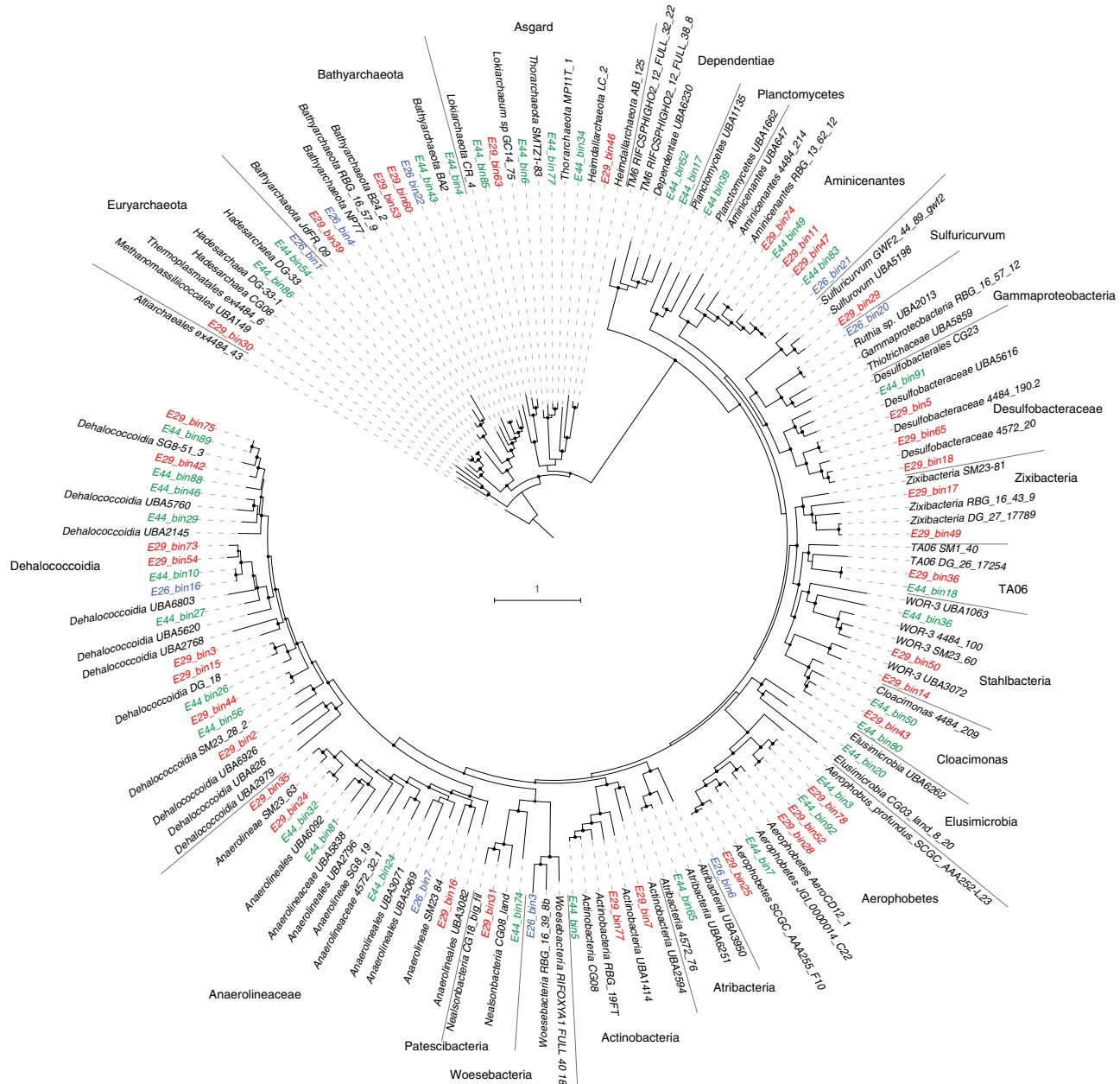

**Fig. 2** Phylogenetic placement of 82 reconstructed metagenome-assembled genomes. A maximum-likelihood phylogenomic tree was built based on concatenated amino acid sequences of 43 conserved single-copy genes using RAxML with the PROTGAMMALG model. Sequences of Altiarchaeales ex4484_43 were used as an outgroup. The scale bar represents 1 amino acid substitution per sequence position. Bootstrap values >70% are indicated. Blue for Site E26 (E26_binX), red for Site E29 (E29_binY), and green for Site E44 (E44_binZ)

matter from the overlying water column or as aliphatic and aromatic petroleum compounds that migrate upwards from underlying petroleum-bearing sediments[8]. With respect to detrital matter, genes involved in carbon acquisition and breakdown were prevalent across both archaeal and bacterial MAGs. These include genes encoding intracellular and extracellular carbohydrate-active enzymes and peptidases, as well as relevant transporters and glycolysis enzymes (Fig. 3 and Supplementary Data 4). The ability to break down fatty acids and other organic acids via the beta-oxidation pathway was identified in 13 MAGs, including members of Chloroflexi, *Deltaproteobacteria*, Aerophobetes and Lokiarchaeota (Fig. 3 and Supplementary Data 4). Metabolomics data supported these genomic predictions and showed a surprising degree of consistency between the geographically distinct sampling sites (Fig. 4). Over 50 metabolites

from eight pathways were detected in all the samples, including carbohydrate metabolism (e.g. glucose), amino acid metabolism (e.g. glutamate), and beta-oxidation (e.g. 10-hydroxydecanoate). Together, the metagenomic and metabolomic data suggest that seabed microorganisms are involved in recycling of residual organic matter, including complex carbohydrates, proteins and lipids.

To identify the potential for microbial degradation of hydrocarbons, we focused on functional marker genes encoding enzymes that initiate anaerobic hydrocarbon biodegradation by activating mechanistically stable C−H bonds[17]. We obtained evidence that two of the four known pathways for oxygen-independent C−H activation[17–20] were present: hydrocarbon addition to fumarate by glycyl-radical enzymes[20] and hydroxylation with water by molybdenum cofactor-containing enzymes[17].

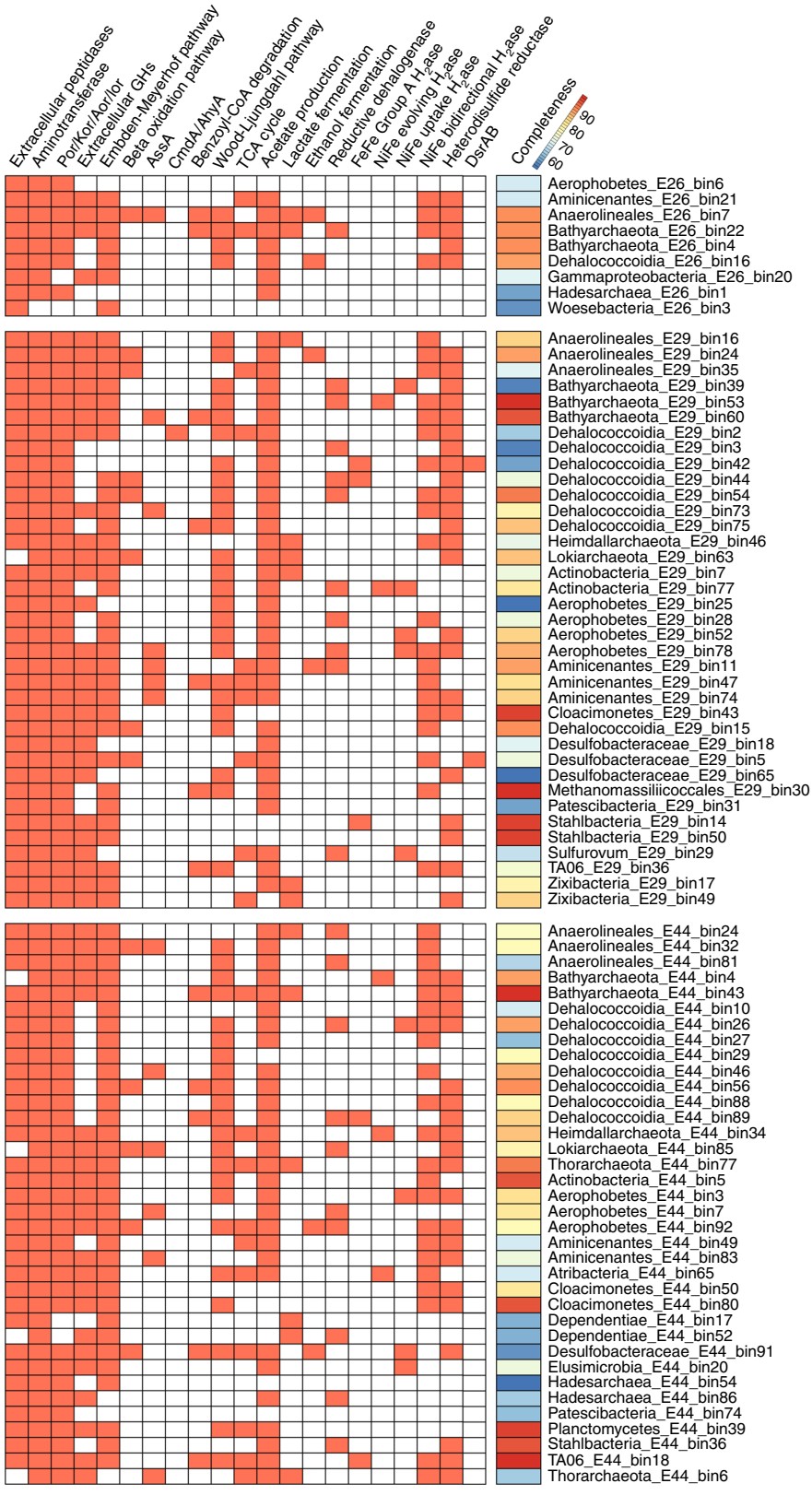

Glycyl-radical enzymes proposed to mediate hydrocarbon addition to fumarate were found in 13 MAGs (Chloroflexi, Aminicenantes, Aerophobetes, Actinobacteria, Bathyarchaeota, Thorarchaeota, and Lokiarchaeota) (Fig. 3). The sequences identified are phylogenetically distant from canonical akyl-/ arylalkylsuccinate synthases, but form a common clade with the glycyl-radical enzymes proposed to mediate alkane activation in anaerobic alkane degraders *Vallitalea guaymasensis* L81 and *Archaeoglobus fulgidus* VC-16 [21–23] (Supplementary Fig. 3). Based on quality-filtered reads, canonical AssA (*n*-alkane succinate synthase) and BssA (benzylsuccinate synthase) enzymes are also encoded at these sites and were most abundant in Site

**Fig. 3** Identification of functional genes or pathways present in metagenome-assembled genomes. The presence of genes or pathways are indicated by orange-shaded boxes. Aor, aldehyde:ferredoxin oxidoreductase; Kor, 2-oxoglutarate/2-oxoacid ferredoxin oxidoreductase; Por, pyruvate:ferredoxin oxidoreductase; Ior, indolepyruvate ferredoxin oxidoreductase; GHs, glycoside hydrolases; AssA, catalytic subunit of alkylsuccinate synthase; CmdA, catalytic subunit of p-cymene dehydrogenase; AhyA, catalytic subunit of alkane $C_2$-methylene hydroxylase; $H_2$ase, hydrogenase; DsrAB, dissimilatory sulfite reductase. Pathways were indicated as being present if at least five genes in the Embden-Meyerhof-Parnas pathway, three genes in the beta-oxidation pathway, four genes in the Wood−Ljungdahl pathway, and six genes in the TCA cycle were detected. Additional details for the central benzoyl-CoA degradation pathway can be found in Supplementary Fig. 5. Lactate and ethanol fermentation are indicated if genes encoding respective dehydrogenases were detected. More details about these functional genes and pathways can be found in the text and in Supplementary Data 4

E29 (Supplementary Table 3). In agreement with this, metabolomics analysis detected six succinic acid conjugates involved in hydrocarbon activation, including conjugates of xylene, toluene, cyclohexane, and propane (Fig. 4). For the hydroxylation pathway, a *Dehalococcoidia* MAG in Site E29 encoded proteins sharing over 40% sequence identities to the catalytic subunits of *p*-cymene dehydrogenase (Cmd) and alkane $C_2$-methylene hydroxylase (Ahy) (Fig. 3 and Supplementary Fig. 4)[24]. Genes encoding enzymes catalyzing hydrocarbon carboxylation (*ubiD*-like), reverse methanogenesis (*mcrA*-like), and aerobic hydrocarbon degradation (e.g. *alkB*) were not detected (Supplementary Table 3). The latter result is expected due to the low concentrations of oxygen in the top 20 cm of organic rich seabed sediments[8].

Our results also provide evidence that aromatic compounds can be anaerobically degraded via channeling into the central benzoyl-CoA degradation pathway. Through metabolomic analysis, we detected multiple intermediates (Fig. 4) involved in both the production and degradation of benzoyl-CoA, a universal intermediate formed during the degradation of aromatic compounds[25]. Various compounds that can be activated to form benzoyl-CoA were detected, including benzoate, benzylsuccinate, 4-hydroxybenzoate, phenylacetate, acetophenone, and phenol. The downstream metabolite glutarate was also highly abundant (Fig. 4). Benzoyl-CoA can be reduced to cyclohex-1,5-diene-1-carboxyl-CoA by either the Class I ATP-dependent benzoyl-CoA reductase pathway (*bcr* genes, e.g. *Thauera aromatica*) or Class II ATP-independent reductase (*bam* genes, e.g. sulfate reducers)[26]. We identified *bcr* genes for the Class I pathways in 12 MAGs from both bacteria (i.e., *Dehalococcoidia*, *Anaerolineae*, *Deltaproteobacteria*, Aminicenantes and TA06) and archaea (i.e., *Thermoplasmata* and *Bathyarchaeota*) (Fig. 3 and Supplementary Fig. 5). Genes for further transformation of dienoyl-CoA to 3-hydroxypimelyl-CoA were also identified, i.e. those encoding 6-oxo-cyclohex-1-ene-carbonyl-CoA hydrolase (*oah*), cyclohex-1,5-diencarbonyl-CoA hydratase (*dch*) and 6-hydroxycyclohex-1-ene-1-carbonyl-CoA dehydrogenases (*had*)[27] (Supplementary Fig. 5). In combination, these results strongly suggest that the organisms represented by these MAGs mediate the typical downstream degradation of aromatic compounds through the central benzoyl-CoA Bcr-Dch-Had-Oah pathway. However, sources of benzoate other than via anaerobic degradation of the above compounds cannot be ruled out based on current data.

**Production and consumption of acetate and hydrogen.** Analysis of MAGs from these deep-sea hydrocarbon-associated sediments suggests that fermentation, rather than respiration, is the primary mode of organic carbon turnover in these environments. Most recovered MAGs with capacity for heterotrophic carbon degradation lacked respiratory primary dehydrogenases and terminal reductases, with exception of several Proteobacteria and one Chloroflexi (Supplementary Data 4). In contrast, various MAGs contained genes indicating the capability for fermentative production of acetate (69 MAGs), lactate (14 MAGs), and ethanol

(6 MAGs) (Fig. 3 and Supplementary Data 4). These findings therefore provide genomic evidence supporting other studies emphasizing the importance of fermentation, including acetate production, in deep-sea sediments[9]. Acetate can also be produced by acetogenic $CO_2$ reduction through the Wood−Ljungdahl pathway using a range of inorganic and organic substrates[10]. Partial or complete sets of genes for the Wood−Ljungdahl pathway were found in 50 MAGs (Fig. 3 and Supplementary Fig. 6), including those affiliated with phyla previously inferred to mediate acetogenesis in deep-sea sediments through either the tetrahydrofolate-dependent bacterial pathway (e.g. Chloroflexi and Aerophobetes)[5,28] or the tetrahydromethanopterin-dependent archaeal variant (e.g. Bathyarchaeota and Asgard group)[29,30]. In addition, the signature diagnostic gene for the Wood−Ljungdahl pathway (*acsB*; acetyl-CoA synthase) is in high relative abundance in the quality-filtered metagenome reads at all three sites (Supplementary Table 3). The most abundant MAG at each site were all putative acetogenic heterotrophs, i.e. *Dehalococcoidia* E26_bin16, Actinobacteria E44_bin5, and Aminicenantes E29_bin47 for Sites E26, E44 and E29 respectively (~3.3–4.5% relative abundance, Supplementary Data 2). These observations are in agreement with mounting evidence that homoacetogens play a quantitatively important role in organic carbon cycling in the marine deep biosphere[29,31,32].

The potential for $H_2$ metabolism was also found in MAGs from all three sites. We screened putative hydrogenase genes from various subgroups in MAGs as well as unbinned metagenomic sequences (Figs. 1 and 3, Supplementary Table 3, and Supplementary Data 4). Surprisingly few $H_2$ evolving-only hydrogenases were observed, with only five Group A [FeFe]-hydrogenases and five Group 4 [NiFe]-hydrogenases detected across the bacterial and archaeal MAGs. Instead, the most abundant hydrogenases within the MAGs and quality-filtered unassembled reads were the Group 3b, 3c, and 3d [NiFe]-hydrogenases. Group 3b and 3d hydrogenases are physiologically reversible, but generally support fermentation in anoxic environments by coupling NAD(P)H reoxidation to fermentative $H_2$ evolution[33]. Group 3c hydrogenases mediate a central step in hydrogenotrophic methanogenesis, bifurcating electrons from $H_2$ to heterodisulfides and ferredoxin[34]; their functional role in bacteria and nonmethanogenic archaea remains unresolved[33] yet corresponding genes frequently co-occur with heterodisulfide reductases across multiple archaeal and bacterial MAGs (Fig. 3). Various Group 1 [NiFe]-hydrogenases were also detected, which are known to support hydrogenotrophic respiration in conjunction with a wide range of terminal reductases. This is consistent with previous studies in the Gulf of Mexico that experimentally measured the potential for hydrogen oxidation catalyzed by hydrogenase enzymes[35].

Given the genomic evidence for hydrogen and acetate production in these sediments, we investigated whether any of the MAGs encoded terminal reductases to respire these compounds. In agreement with porewater sulfate concentrations (16–27 mM; see Supplementary Note 2), the key genes for dissimilatory sulfate reduction (*dsrAB*) were present across the

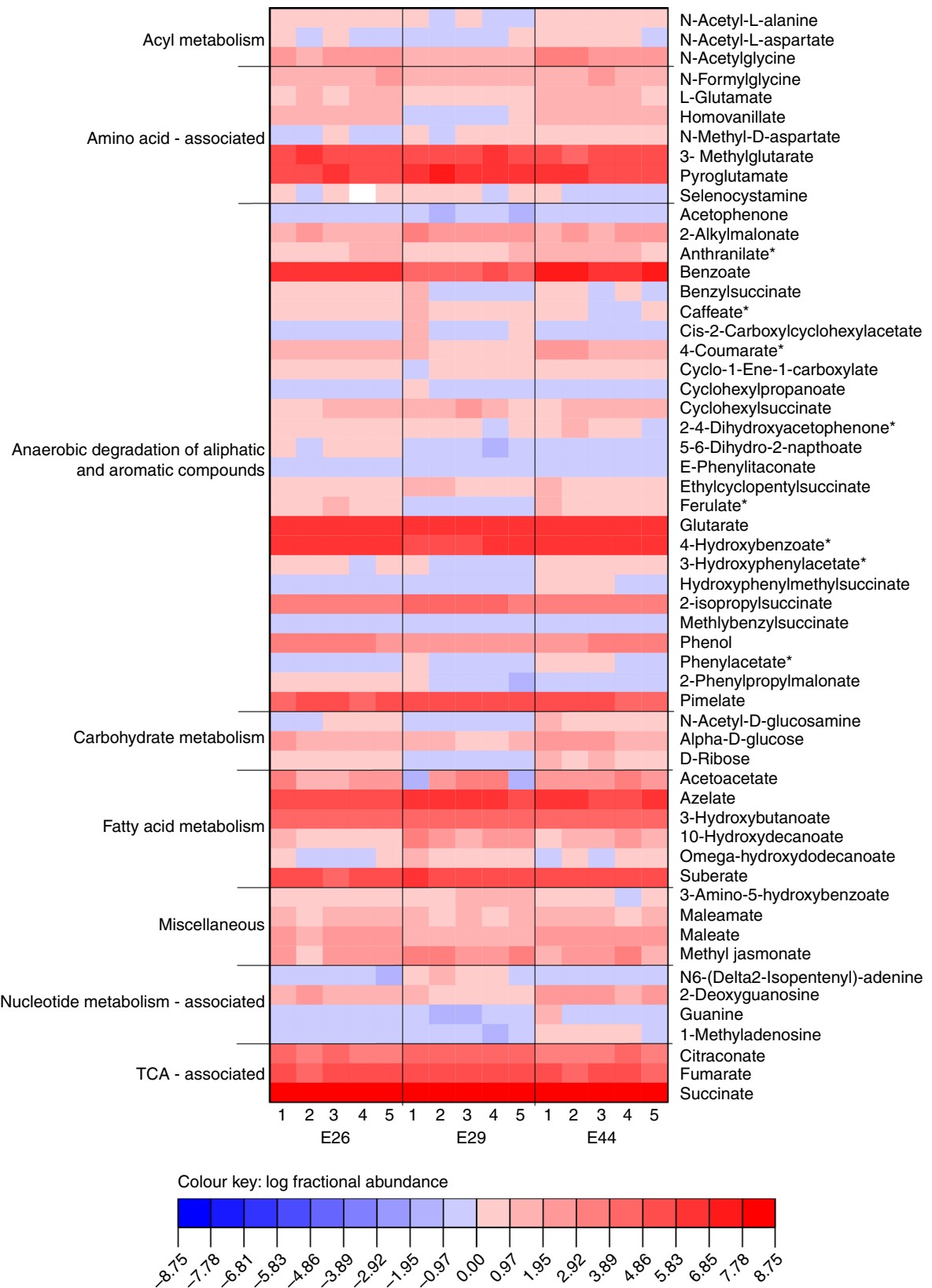

**Fig. 4** Heatmap of metabolites identified in sediment pore water at Sites E26, E29, and E44. Metabolite levels were measured by LC-MS and observed intensities are expressed as the log fractional abundance. Technical replicate numbers ($n = 5$) from each site are given on the bottom axis. Compound names are listed on the right axis and pathway assignments on the left. Compounds denoted with an asterisk can be intermediates in both anaerobic and aerobic metabolic pathways

**Table 2 Thermodynamic parameters and Gibbs free energies for anaerobic benzoate and hexadecane degradation scenarios**

| Substrates | Reaction types | Reactions | $\Delta G^o$ (kJ) | $\Delta H^o$ (kJ) | $\Delta G$ (kJ)[a] |
|---|---|---|---|---|---|
| Hexadecane | 1. Hydrogenogenic oxidation | $C_{16}H_{34} + 16H_2O \rightarrow 8CH_3COO^- + 17H_2 + 8\,H^+$ | 1089.1 | 1069.8 | 753.3 |
| | 2. Acetogenic oxidation | $C_{16}H_{34} + 8.5HCO_3^- \rightarrow 12.25CH_3COO^- + 3.75\,H^+ + H_2O$ | 176.3 | 98.4 | 133.9 |
| | 3. Complete oxidation | $C_{16}H_{34} + 12.25SO_4^{2-} \rightarrow 16HCO_3^- + 12.25HS^- + H_2O + 3.75\,H^+$ | −407.3 | 24.9 | −653.3 |
| Benzoate | 4. Hydrogenogenic oxidation | $C_7H_5O_2^- + 7H_2O \rightarrow 3CH_3COO^- + HCO_3^- + 3\,H_2 + 3H^+$ | 120.9 | 117.4 | 12.8 |
| | 5. Acetogenic oxidation | $C_7H_5O_2^- + 0.5HCO_3^- \rightarrow 3.75CH_3COO^- + 2.25\,H^+$ | −846.3 | −1091.3 | −950.3 |
| | 6. Complete oxidation | $C_7H_5O_2^- + 3.75SO_4^{2-} + 4H_2O \rightarrow 7HCO_3^- + 3.755HS^- + 2.25\,H^+$ | −76.2 | 29.4 | −230.7 |

[a]The Gibbs free energies were calculated for deep-sea conditions of 4 °C, 300 atm, pH 8 and 2 mM bicarbonate concentrations[31]

metagenome reads, particularly at Site E29 (Supplementary Table 3); however, probably due to incompleteness of genomes or insufficient binning, these genes were identified only in two MAGs affiliated with *Deltaproteobacteria* and *Dehalococcoidia* (Supplementary Data 4). We also identified 31 putative reductive dehalogenase genes (*rdhA*) across 22 MAGs, mainly from *Aminicenantes* and *Bathyarchaeota* (Fig. 3 and Supplementary Data 4); this suggests that organohalides, which can be produced through abiotic and biotic processes in marine ecosystems[36], may be electron acceptors in these deep-sea sediments. At least two thirds of the MAGs corresponding to putative sulfate reducers and dehalorespirers encoded the capacity to completely oxidize acetate and other organic acids to $CO_2$ using either the reverse Wood−Ljungdahl pathway or TCA cycle (Fig. 3 and Supplementary Data 4). Several of these MAGs also harbored the capacity for hydrogenotrophic dehalorespiration via Group 1a and 1b [NiFe]-hydrogenases (Fig. 3). In addition to these dominant uptake pathways, one MAG belonging to the epsilonproteobacterial genus *Sulfurovum* (E29_bin29) included genes for the enzymes needed to oxidize either $H_2$ (group 1b [NiFe]-hydrogenase), elemental sulfur (*soxABXYZ*), and sulfide (*sqr*), using nitrate as an electron acceptor (*napAGH*); this MAG also has a complete set of genes for autotrophic $CO_2$ fixation via the reductive TCA cycle (Fig. 3 and Supplementary Data 4).

The capacity for methanogenesis appears to be relatively low. The genes for methanogenesis were detected in quality-filtered unassembled reads in all three sediments and were mainly affiliated with acetoclastic methanogens at Site E29, and hydrogenotrophic methanogens at the other two sites (Fig. 1d). However, none of the MAGs contained *mcrA* genes. Overall, the collectively weak *mcrA* signal in the metagenomes suggests that the high levels of biogenic methane detected by geochemical analysis (Table 1) is due to methanogenesis in deeper sediment layers. Similar phenomena have been observed in other sites, where *mcrA* genes are in low abundance despite clear geochemical evidence for biogenic methane[37]. Sequencing additional sediment depths at greater resolution would likely result in detection of methanogens and ANME lineages harboring *mcrA* and related genes.

**Thermodynamic modeling**. Together, the geochemical, metabolomic, and metagenomic data strongly indicate that anaerobic degradation of aliphatic and aromatic compounds occurs in these deep-sea sediments (Table 1; Figs. 3 and 4). Recreating the environmental conditions for cultivating the organisms represented by the retrieved MAGs is a challenging process, preventing further validation of the degradation capabilities of these compounds (and other metabolisms) among the majority of the lineages represented by the MAGs retrieved here[32]. Instead, we provide theoretical evidence that anaerobic degradation of aliphatic and aromatic compounds is feasible in this environment by modeling whether these processes are thermodynamically favorable in the conditions typical of deep-sea sediments, namely high pressure and low temperature.

As concluded from the genome analysis and supported by metabolomics (Figs. 3 and 4), it is likely that anaerobic degradation occurs through an incomplete oxidation pathway. However, due to incompleteness of the reconstructed genomes, we cannot exclude the possibility that complete oxidation of aliphatic and aromatic compounds to $CO_2$ occurs through, for example, coupling with sulfate reduction (Fig. 3). Additionally, several recent studies indicate that some aliphatic and aromatic compounds can be incompletely oxidized to acetate via the Wood−Ljungdahl pathway[28,31,38]. Therefore, we compared the thermodynamic constraints on anaerobic biodegradation for three plausible scenarios (Table 2): (i) incomplete oxidation with production of hydrogen and acetate, (ii) acetogenic oxidation with production of acetate alone, and (iii) complete oxidation coupled with sulfate reduction. Hexadecane and benzoate are used as representative aliphatic and aromatic compounds, respectively, based on the results of geochemistry and metabolomics results (e.g. $C_{2+}$ alkane and benzoate detection) and genomic analysis (e.g. genes encoding for glycyl-radical enzymes and *bcr* genes). Under deep-sea conditions, without taking the in situ concentrations of hydrogen and acetate into consideration, the calculations show that sulfate-dependent complete oxidation of hexadecane and benzoate, as well as acetogenic oxidation of benzoate, would be energetically favorable. The three other reactions would be endergonic, but based on the measured concentrations for both acetate and $H_2$ in these sediments being low (Supplementary Note 2) these reactions could also take place in theory (Table 2 and Fig. 5). This suggests that acetate- and $H_2$-scavengers, by making acetogenic and hydrogenogenic degradation more thermodynamically favorable, may support activity of anaerobic degraders in the community.

## Discussion

In this study, metagenomics revealed that most of the bacteria and archaea in the deep-sea sediment microbial communities sampled belong to candidate phyla that lack cultured representatives and sequenced genomes (Figs. 1 and 2). As a consequence, it is challenging to link phylogenetic patterns with the microbial functional traits underpinning the biogeochemistry of deep seabed habitats. Here, we were able to address this by combining de novo assembly and binning of metagenomic data with petroleum geochemistry, metabolite identification, and thermodynamic modeling. Pathway reconstruction from 82 MAGs recovered from the three deep-sea near-surface sediments revealed that many community members were capable of acquiring and hydrolyzing residual organic matter (Fig. 3), whether supplied as detritus from the overlying water column or as autochthonously produced necromass. Heterotrophic fermenters and acetogens were in considerably higher relative abundance than heterotrophic respirers, despite the abundance of sulfate in the sediments (Supplementary Note 2). For example, while

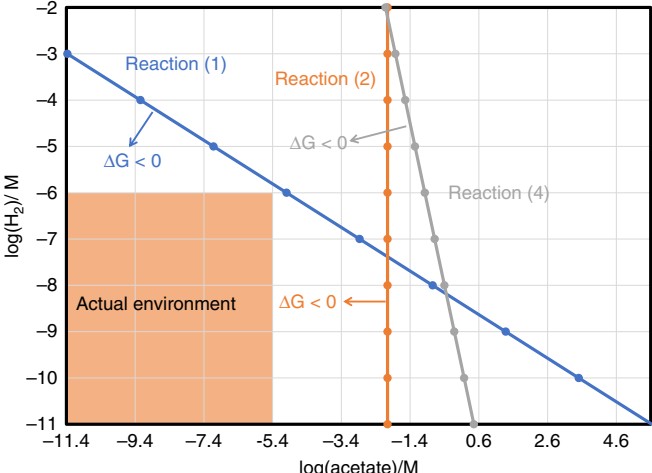

**Fig. 5** Thermodynamic constraints on anaerobic benzoate and hexadecane degradation. Three reactions (Reactions 1, 2, and 4) are illustrated as taken from Table 2 where ΔG > 0. Thermodynamics for each reaction are indicated by a line in its corresponding color. If ΔG < 0, the reaction is energetically favorable (indicated by arrow), and if ΔG > 0 the reaction is assumed not to occur. In the studied environment (outlined in the shaded area), hydrogen concentrations fall below 1 μM while acetate concentrations are below 2.5 μM. The graph shows that ΔG for three reactions are all negative when both actual concentrations of acetate and hydrogen are taken into consideration

genomic coverage of putative sulfate reducers is relatively low (<1% of the communities), putative acetogenic heterotrophs were the most abundant community members at each site. Therefore, microbial communities in the deep seabed are likely shaped more by the capacity to utilize available electron donors than by the availability of oxidants. In line with the different geochemical profiles at the three sites (Table 1), some differences in the composition of microbial communities and the abundance of key metabolic genes were observed (Fig. 1 and Supplementary Table 3). However, metabolic capabilities such as fermentation, acetogenesis, and hydrogen metabolism were conserved across diverse phyla in each site (Fig. 3). This suggests some functional redundancy in these microbial communities, similar to that recently inferred in a study of Guaymas Basin hydrothermal sediments[39].

In this context, multiple lines of evidence indicate aliphatic or aromatic compounds serve as carbon and energy sources for anaerobic populations in these deep-sea hydrocarbon seep environments (Tables 1, 2; Figs. 3–5). Whereas capacity for detrital organic matter degradation is a common feature in the genomes retrieved in this study, and from many other environments in general[40], anaerobic degradation of aliphatic or aromatic compounds is a more exclusive feature that was detected in 19 out of 82 MAGs. In all three sediments, there was metagenomic and metabolomic evidence for anaerobic hydrocarbon oxidation via hydrocarbon addition to fumarate and hydroxylation pathways, as well as anaerobic aromatic compound degradation by the Class I benzoyl-CoA reductase pathway. The ability to utilize aliphatic or aromatic compounds may explain the ecological dominance (high relative abundance) of certain lineages of bacteria and archaea in these microbial communities (Fig. 1a). Many of the detected phyla have previously been found to be associated with hydrocarbons in various settings. For example, Aerophobetes have been detected in other cold seep environments[5], Aminicenantes are often found associated with fossil fuels[41], and Chloroflexi harboring genes for anaerobic hydrocarbon degradation have been found in hydrothermal vent sediments[2]. While archaea

have been reported to mediate oxidation of methane and other short-chain alkanes in sediments[3,18], few have been reported to anaerobically degrade larger hydrocarbons[24]. The finding that Bathyarchaeota and other archaeal phyla are potentially capable of anaerobic degradation of aliphatic or aromatic compounds extends the potential substrate spectrum for archaea. More broadly, building on recent findings[18], this work emphasize that nonmethane aliphatic and aromatic compounds could significantly contribute to carbon and energy budgets in these deep-sea settings.

Genomic analyses of the 12 MAGs harboring genes for central benzoyl-CoA pathway indicate they have an acetogenic lifestyle. The finding that these organisms use the ATP-consuming class I, not the reversible class II, benzoyl-CoA reductase is surprising. It is generally thought that strict anaerobes must use class II BCRs because the amount of energy available from benzoate oxidation during sulfate reduction or fermentation is not sufficient to support the substantial energetic requirement of the ATP-dependent class I BCR reaction[27]. However, there are reported exceptions to these observations, such as the hyperthermophilic archaeon *Ferroglobus placidus* that couples benzoate degradation via the Class I system with iron reduction[27], and fermentative deep-sea Chloroflexi strains DscP3 and Dsc4 that contain genes for class I benzoyl-CoA reductases[28]. Acetogenic oxidation may explain why the Class I reaction is energetically favorable; thermodynamic modeling indicates the free Gibbs energy yield of acetogenic oxidation of benzoate is much higher than its hydrogenogenic or complete oxidation (Table 2). These inferences are in line with mounting evidence that acetogens are important community members in energy-limited seafloor ecosystems[29,31]. Acetogens may be favored in such ecosystems given their utilization of many organic compounds is thermodynamically favorable, and their relatively high ATP yields[29,31].

Based on the evidence presented here, we propose that acetate and hydrogen are the central intermediates underpinning community interactions and biogeochemical cycling in these deep-sea sediments. Despite the presence of putative acetogens and hydrogenogens, acetate and hydrogen were below the limits of detection in sediment porewater (Supplementary Note 2), indicating both compounds are rapidly turned over by other community members. Consistently, microbial communities encoded the genes for the coupling of acetate consumption to sulfate reduction, organohalide respiration, and acetoclastic methanogenesis, consistent with other studies[38,42]. Some community members also appear to be capable of $H_2$ consumption, including through respiratory membrane-bound enzymes and reversible cytosolic enzymes. In turn, hydrogen oxidation can support autotrophic carbon fixation and therefore may provide a feedback loop for regeneration of organic carbon. Moreover, acetate- and hydrogen-oxidizing community members are likely to promote upstream acetogenic and hydrogenogenic degradation of necromass and aliphatic or aromatic compounds. Thermodynamic modeling indicates that maintaining low acetate and hydrogen concentrations in the environment is important for promoting continuous oxidation of organic substrates (Table 2 and Fig. 5).

## Methods

**Sample selection based on geochemical characterization**. The three marine sediment samples used in this study were chosen from among several sites sampled as part of a piston coring seafloor survey in the Eastern Gulf of Mexico, as described previously[13]. Piston cores penetrating 5−6 m below seafloor (mbsf) were sectioned in 20 cm intervals on board the research vessel immediately following their retrieval. Three intervals from the bottom half of the core were chosen for geochemical analysis, and were either frozen immediately (for liquid hydrocarbon analyses), or flushed with $N_2$ and sealed in hydrocarbon-free gas tight metal canisters then frozen until analysis (for gaseous hydrocarbon analysis). Interstitial gas analysis was later performed on the headspace in the canisters using GC

with Flame Ionization Detector (GC-FID). Sediment samples for gas/liquid chromatography and stable isotope analysis were frozen, freeze-dried and homogenized then extracted using accelerated solvent extraction (ACE 200). Extracts were subsequently analyzed using GC/FID, GC/MS, a Perkin-Elmer Model LS 50B fluorometer, and Finnigan MAT 252 isotope mass spectrometry as detailed elsewhere[43]. On the basis of TSF and UCM concentration thresholds described previously[13], core segments from E26 and E29 were qualified and core segments from E44 were disqualified for unambiguous occurrence of thermogenic liquid hydrocarbons. Additionally, interstitial hydrocarbon gases were observed in the core segments of E29. Samples from the surface 0–20 cm interval from these three cores were further analyzed as described below.

**Porewater geochemistry.** Porewater sulfate and chloride concentrations were measured in a Dionex ICS-5000 reagent-free ion chromatography system (Thermo Scientific, CA, USA) equipped with an anion-exchange column (Dionex IonPac AS22; $4 \times 250$ mm; Thermo Scientific), an EGC-500 $K_2CO_3$ eluent generator cartridge and a conductivity detector. Organic acids were analyzed in the 0.2 μm filtered sediment porewater using a Thermo RS3000 HPLC fitted with an Ultimate 3000 UV detector. Separation was achieved over an Aminex HPX-87H organic acid column (BioRad, USA) under isocratic conditions (0.05 mM $H_2SO_4$) at 60 °C with a run time of 20 min. Organic acids were compared to the retention time of known standards and the limit of detection for acetate was determined to be 2.5 μM.

**Metabolomic analysis.** For the analysis of metabolites, sediment was centrifuged at $21,100 \times g$ for 10 min at room temperature, the supernatant was collected, diluted 1:1 in pure methanol, and filtered through 0.2 μm Teflon syringe filters. Each sediment was subsampled five times to assess technical variability across the sample. Metabolites present in the extracts were separated using ultra high-performance liquid chromatography (UHPLC) performed using a gradient of 20 mM ammonium formate at pH 3.0 in water (solvent A) and 0.1% formic acid (% v/v) in acetonitrile (solvent B) in conjunction with a Syncronis™ HILIC LC column (100 mm × 2.1 mm × 2.1 μm; Thermo Scientific). High-resolution mass spectral data were acquired on a Thermo Scientific Q-Exactive™ HF Hybrid Quadrupole-Orbitrap mass spectrometer coupled to an electrospray ionization source. Data were acquired in negative ion full-scan mode from 50–750 m/z at 240,000 resolution with an automatic gain control (AGC) target of 3e6 and a maximum injection time of 200 ms. For MS/MS fragmentation experiments, an isolation window of 1 m/z and an AGC target of 1e6 was used with a maximum injection time of 100 ms. Data were analyzed in MAVEN[44]. Metabolites were assigned based on accurate mass and retention times of observed signals relative to standards (where available). Metabolites classified as being involved in the anaerobic degradation of aliphatic and aromatic compound pathways[45], for which metabolites standards are not readily available, were assigned using accurate mass alone. The key benzoate and succinate metabolites were assigned using accurate mass, co-elution and MS/MS fragmentation patterns. To control for variability in total organic content across the sediment samples, metabolite data are presented based on their fractional abundance relative to all observed metabolites (i.e. constant sum normalization) and were visualized based on their log fractional abundance[46].

**DNA extraction and sequencing.** For the three sediment samples, DNA was extracted from 10 g of sediment using the PowerMax Soil DNA Isolation Kit (12988-10, QIAGEN) according to the manufacturer's protocol with minor modifications for the step of homogenization and cell lysis, i.e., cells were lysed in PowerMax Bead Solution tubes for 45 s at 5.5 m s$^{-1}$ using a Bead Ruptor 24 (OMNI International). DNA concentrations were assessed using a Qubit 2.0 fluorometer (Thermo Fisher Scientific, Canada). Metagenomic library preparation and DNA sequencing was conducted at the Center for Health Genomics and Informatics in the Cumming School of Medicine, University of Calgary. DNA fragment libraries were prepared by shearing genomic DNA using a Covaris sonicator and the NEBNext Ultra II DNA library preparation kit (New England BioLabs). DNA was sequenced on a ~40 Gb (i.e. 130 M reads) mid-output NextSeq 500 System (Illumina Inc.) 300 cycle (2 × 150 bp) sequencing run.

To provide a high-resolution microbial community profile, as well as quantitative insights into microbial community diversity, the three samples were also subjected to 16S rRNA gene amplicon sequencing on a MiSeq benchtop sequencer (Illumina Inc.). DNA was extracted from separate aliquots of the same sediment samples using the DNeasy PowerLyzer PowerSoil kit (MO BIO Laboratories, a Qiagen Company, Carlsbad, CA, USA) and used as the template for different PCR reactions. The v3–4 region of the bacterial 16S rRNA gene and the v4–5 region of the archaeal 16S rRNA gene were amplified using the primer pairs SD-Bact-0341-bS17/SD-Bact-0785-aA21 and SD-Arch-0519-aS15/SD-Arch-0911-aA20, respectively[47] on a ~15 Gb 600-cycle (2 × 300 bp) sequencing run.

**Metagenomic assembly and binning.** Raw reads were quality-controlled by (1) clipping off primers and adapters and (2) filtering out artifacts and low-quality reads as described previously[48]. Filtered reads were assembled using metaSPAdes version 3.11.0 [49] and short contigs (<500 bp) were removed. Sequence coverage was determined by mapping filtered reads onto assembled contigs using BBmap version

36 (https://sourceforge.net/projects/bbmap/). Binning of metagenome contigs was performed using MetaBAT version 2.12.1 (–minContig 1500)[50]. Contaminated contigs in the produced bins were further removed based on genomic properties (GC, tetranucleotide signatures, and coverage) and taxonomic assignments using RefineM version 0.0.22 [51]. Resulting bins were further examined for contamination and completeness using CheckM version 1.0.8 with the lineage-specific workflow[16].

**Annotation.** For MAGs, genes were called by Prodigal (-p meta)[52]. Metabolic pathways were predicted against the KEGG GENES database using the Ghost-KOALA tool[53] and against the Pfam, TIGRfam and custom HMM databases (https://github.com/banfieldlab/metabolic-hmms) using MetaErg (https://sourceforge.net/projects/metaerg/). The dbCAN web server was used for carbohydrate-active gene identification (cutoffs: coverage fraction: 0.40; e-value: 1e-18)[54]. Genes encoding proteases and peptidases were identified using BLASTp against the MEROPS database release 12.0 (cutoffs: e-value, 1e-20; sequence identity, 30%)[55]. Genes involved in anaerobic hydrocarbon degradation were identified using BLASTp against a custom database (Supplementary Data 5) (cutoffs: e-value, 1e-20; sequence identity, 30%). Hydrogenases were identified and classified using a web-based search using the hydrogenase classifier HydDB[56].

Full-length 16S rRNA genes were reconstructed from metagenomic reads using phyloFlash version 3.1 (https://hrgv.github.io/phyloFlash/) together with the SILVA SSU 132 rRNA database[57]. Diversity calculations were based on separate 16S rRNA gene amplicon library results[13]. Functional and taxonomic McrA gpkgs were used to assess the diversity of methanogens against the metagenomic reads using GraftM with default parameters[58]. Genes encoding the catalytic subunits of hydrogenases, *dsrA*, *acsB*, *assA*, *nmsA* and *bssA* were retrieved from metagenomic reads through diamond BLASTx[59] queries against comprehensive custom databases[30,56] (cutoffs: e-value, 1e-10; sequence identity, 70%).

**Phylogenetic analyses.** For taxonomic classification of each MAG, two methods were used to produce genome trees that were then used to validate each other. In the first method the tree was constructed using concatenated proteins of up to 16 syntenic ribosomal protein genes following procedures reported elsewhere[60]; the second tree was constructed using concatenated amino acid sequences of up to 43 conserved single-copy genes following procedures described previously[61]. Both trees were calculated using FastTree version 2.1.9 (-lg -gamma)[62] and resulting phylogenies were congruent. Reference genomes for relatives were accessed from NCBI GenBank, including genomes selected from several recent studies representing the majority of candidate bacterial and archaeal phylogenetic groups[2,51,63,64]. The tree in Fig. 2 was inferred based on concatenation of 43 conserved single-copy genes (Supplementary Data 1). Specifically, it was built using RAxML version 8 [65] implemented by the CIPRES Science Gateway[66] and it was called as follows: raxmlHPC-HYBRID -f a -n result -s input -c 25 -N 100 -p 12345 -m PROTCATLG -x 12345. The phylogeny resulting from RAxML is consistent with the taxonomic classification of MAGs that resulted from FastTree. Interactive tree of life (iTOL) version 3 [67] was used for tree visualization and modification.

For phylogenetic placements of functional genes, sequences were aligned using the ClustalW algorithm included in MEGA7 [68]. All positions with less than 95% site coverage were eliminated. Maximum-likelihood phylogenetic trees were constructed in MEGA7 and evolutionary distances were computed using the Poisson correction method. These trees were bootstrapped with 50 replicates.

**Thermodynamic calculations.** The values of Gibbs free energy of formation for substances were taken from Madigan et al.[69] and Dolfing et al.[42]. The pH used in all calculations was 8.0 as reported in a previous thermodynamic study of deep buried sediments[31], partial pressure was 300 atm based on water depths at the three sites (http://docs.bluerobotics.com/calc/pressure-depth/), and temperature was set as 4 °C to represent deep-sea conditions. Calculations and corrections based on actual temperatures, pressure, and concentrations followed accepted protocols for determining reaction kinetics and thermodynamics[70].

**Reporting summary.** Further information on research design is available in the Nature Research Reporting Summary linked to this article.

## Data availability

DNA sequences (amplicon sequences, genomes and raw sequence reads) have been deposited in the NCBI BioProject database with accession codes PRJNA415828 and PRJNA485648. Individual assembly for metagenome-assembled genomes can also be found at figshare (https://figshare.com/s/9570b8a8ff818bbb0c8f). Genome sequences used to determine the phylogeny in Fig. 2 can be found at figshare (https://figshare.com/s/355963dc21a263e34c1f). All other data supporting the findings of this study are available within the article and its supplementary information files, or from the corresponding authors upon request.

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

## Acknowledgements

The work was supported by Genome Canada Genomics Applications Partnership Program (GAPP) and Canada Foundation for Innovation (CFI-JELF 33752) awards to C.R.J.H., who is supported by a Campus Alberta Innovates Program Chair. Metabolomics data were acquired by R.A.G. at the Calgary Metabolomics Research Facility (CMRF), which is supported by the International Microbiome Centre and the Canada Foundation for Innovation (CFI-JELF 34986). I.A.L. is supported by an Alberta Innovates Translational Health Chair. C.G. is supported by an ARC DECRA Fellowship (DE170100310) and an ARC Discovery Project (DP180101762). We thank Xiaoli Dong and Marc Strous for establishing bioinformatics workflows and pipelines, the Centre for Health Genomics and Informatics at University of Calgary for NextSeq sequencing, Alexander Probst for providing the database of 16 syntenic ribosomal proteins, Nina Dombrowski and Brett Baker for providing a custom blast database for hydrocarbon degradation genes, and Rhonda Clark for research support.

## Author contributions

X.D. and C.R.J.H. designed the study. X.D. and C.G. processed the data, reconstructed the genomes and performed the genome analyses. X.D., J.D. and D.M. performed the thermodynamics analysis. M.C. confirmed phylogenetic analyses of genomes. A.C. and C.L. conducted amplicon sequencing and analyses. J.M.B. and B.B.B. collected samples and performed petroleum geochemistry analyses. J.E.R., R.A.G. and I.A.L. performed metabolomics analyses and data interpretation. X.D., C.G. and C.R.J.H. drafted the manuscript. All authors reviewed the results and participated in the writing of the manuscript.

## Additional information

**Competing interests:** The authors declare no competing interests.

