## [Peer Review File · Nature Communications]

Reviewers' comments:

Reviewer #1 (Remarks to the Author):

In this work, the authors conducted massive metagenome sequencing on three sediment samples from hydrocarbon-impacted deep sea sediments in the Gulf of Mexico. The upper 20 cm sediment was investigated. In total, 82 genomes were binned from the metagenome data, which is an excellent outcome considering the high diversity of marine sediments. Identification of hydrocarbon-activating enzymes was pursued not only by following automated annotation pipelines, but also by direct search using known enzymes, which adds to the robustness of the analysis. The work was complemented by sediment geochemistry, diversity based on amplicon sequencing, and metabolomics. Reconstruction of metabolic potential of those sediments from the data resulted in a complex metabolic network, including oxidation of detrital biomass, hydrocarbons, aromatic compounds, acetate and hydrogen.

Although the data generated is robust and adds to our knowledge of the metabolic potential of deep sea sediments, the interpretation suffers from inconsistencies, over-interpretation or unsupported conclusions that need to be carefully considered (see below).

Although the work is introduced as focused on anaerobic hydrocarbon oxidation (Title), this is misleading since the rest of the manuscript (from Abstract on) is rather focused on the whole metabolic potential of these sediments. This does not mean the work is less valuable, but the focus should be clearly stated. I recommend changing the title to better reflect the contents, and guide the expectations of the reader.

The study does not appear to have uncovered a very high potential for anaerobic hydrocarbon oxidation. The only proteins directly related to anaerobic hydrocarbon activation are the divergent AssA, for alkane activation. This is supported to some extent by metabolomics, where some alkyl-succinates have been found (although the metabolomics data are not very visible in the manuscript).

No direct evidence for the anaerobic oxidation of aromatic hydrocarbons was provided. The enzymes for anaerobic benzoate oxidation cannot be used as indicators of hydrocarbon oxidation: it is true that benzoate is an intermediate in the anaerobic oxidation of many aromatic hydrocarbons, but it can also be formed from other sources. Bss/Nms would provide direct evidence for aromatic hydrocarbon oxidation, but encoding genes were not found.

Considering the above, some statements regarding the potential for anaerobic hydrocarbon oxidation should be toned down (for ex. L205, 336). Also, the oxidation of aromatic compounds should be in a way decoupled from that of aromatic hydrocarbons, or make clear that the finding of Bcr for example is not proof for hydrocarbon oxidation.

'Fermentation' is used in a wrong way when applied to alkane (hexadecane) oxidation (Section Thermodynamic modelling of hydrocarbon degradation). Alkanes are reduced compounds which cannot be fermented. They are indeed partly oxidized to acetate and hydrogen (reduction of protons) in syntrophic reactions (Zengler, K., et al., Methane formation from long-chain alkanes by anaerobic microorganisms. *Nature* 401, 266-269, 1999): the correct term would be partial or incomplete oxidation, or syntrophic oxidation. Benzoate can be fermented, but the products are acetate and cyclohex-1-ene carboxylate (e.g. Elshahed, M. S. & McInerney, M. J. Benzoate Fermentation by the Anaerobic Bacterium *Syntrophus aciditrophicus* in the Absence of Hydrogen-Using Microorganisms. *Appl. Environ. Microbiol.* 67, 5520-5525, 2001). Syntrophic benzoate oxidation would also yield acetate and hydrogen (same reference as above). In view of the above, what is the basis for the proposed reactions (Fig. 5) yielding only acetate and no H₂? What would be the electron acceptor in that case?

Also, the author's conclusion that hydrocarbons are mostly incompletely oxidized (Line 296 and fwd.)

is surprising, and should be reconsidered: almost all MAGs recovered with AssA-like proteins contain (according to Fig. 3) either WL pathway or TCA cycle enzymes (or even both). Both pathways can operate in reverse, and can be used for complete oxidation of acetate to CO₂. Also, it is stated in the section ahead (L262 and fwd.) that genes for sulfate reduction and dehalorespiration have been found across the metagenome reads. Absence of specific pathways for the reduction of a terminal electron acceptor in AssA-containing bins may be due to binning or incomplete genomes, for example.

Conclusion at L309-315 is wrong: how could an energy-demanding CO₂ fixation pathway (WL, CO₂ to acetate) be energetically more favorable than an energy-conserving one (incomplete alkane oxidation), no matter how less energetically favorable is the latter, when both end up in the same acetate pool? See energetics of syntrophic hexadecane oxidation in methanogenic cultures (Zengler, K., et al., Methane formation from long-chain alkanes by anaerobic microorganisms. *Nature* 401, 266-269, 1999).

L355-357: this is an understatement, in view of the amount of literature published in the past decade on anaerobic hydrocarbon degradation; and there are plenty of studies with marine cultures (AOM literature, nonmethane gaseous alkane degraders (e.g. Kniemeyer, O., et al., Anaerobic oxidation of short-chain hydrocarbons by marine sulphate-reducing bacteria. *Nature* 449, 898-U810, 2007), strains AK-01 and Hxd3 (Univ. of Oklahoma publications), etc.

L357-361: there is no basis for these statements, and they should be removed. 'Mechanisms' refers to the biochemistry of hydrocarbon activation, and the present work does not provide hints to novel mechanisms.

Other suggestions for improvement

Overall, the manuscript may benefit from shortening and better streamlining the main ideas. The relevance of some sections is not right away clear, like the section at L136-145, dedicated to candidate phylum TA06.

L100: methylsilyl esters are not metabolites. They are formed by sample derivatization upon extraction.

L183 and elsewhere: Gly-radical enzymes of hydrocarbon activation do not catalyze 'fumarate addition' although this is a phrasing used often. They catalyze the cleavage of the hydrocarbon C-H bond, which then adds to fumarate (which is a co-substrate). The correct reference is 'hydrocarbon addition to fumarate'

L173-174: C-H bonds are not mechanistically sophisticated: they are simple, stable, apolar bonds.

L235: there are no such things as heterotrophic compounds

L314/15: here benzoate and related compounds are listed as hydrocarbons.

Reviewer #2 (Remarks to the Author):

Dong and coauthors report 82 new MAGs from three hydrocarbon-rich surface sediments. They discuss new potential roles for uncultured organisms in hydrocarbon degradation, and the importance of hydrogen and acetate as intermediates. In general, this work seems straightforward and useful. I just have a few points of interest.

The abstract and the methods refer to having performed metabolomics. However, it appears that only benzoate and succinate are listed in Table 1. I don't think it's a problem if many metabolites were measured, but the authors only discuss these two. However, it is confusing to expect an array of metabolites, and then only really have two that are buried in Table 1. I say buried because they seem like they were part of a separate geochemical assay designed to measure each compound, as is true for all the other data in Table 1. I think the authors should discuss how many other metabolites were identified and why they are not analyzed. Also, the succinate should be discussed in the text somewhere.

I was not sure how sulfate could go so high above 28 mM. I'm not aware of any methods to concentrate sulfate in marine sediments – is it possible that the calibration curves were inaccurate? I find it very surprising that no methanogens or *mcrA* were found, when the $\delta^{13}C$ of the methane clearly indicates methanogenesis. I agree with the authors that a likely explanation is that methanogens could be deeper than the samples. However, the samples were in the upper 20 cm, which should be sufficiently deep to avoid oxygen. The lack of aerobic genes supports this. Do the authors know of other instances of petroleum-rich sites where methanogens were absent? They might consider mentioning the possibility that some of these hydrocarbon-degrading pathways that are so abundant produce methane independently of *mcrA*.

Figure 6 seems really uninformative. All the processes mentioned are things that are known to occur in hydrocarbon-rich marine sediments. It seems that the novel addition of this paper is that uncultured groups appear to participate in this process. Perhaps this information could be included in this figure.

The conclusion that deep-sea sediment communities are electron-donor driven and coastal sediments are electron acceptor driven (lines 333-335) seems like a strange distinction. Why would this be the case? Also, it is based on the fact that terminal respiring organisms appear to be in low abundance in these samples, but studies such as Leloup et al. 2009, EMI, suggest that this is also the case in coastal sediments.

In summary, I think that the novel findings in this paper are the potential ability of deeply-branching uncultured clades to degrade hydrocarbons at deep-sea seeps. However, these findings are buried among much more general statements. For instance, the title says nothing about this. We have long known that anaerobic hydrocarbon degradation occurs in deep-sea sediments. I suggest that the authors bring their novel discoveries to the forefront, starting with the title.

Reviewer #3 (Remarks to the Author):

A) Key results:

The authors used multiple sources of information to investigate microbial processes and carbon cycling in deep seawater sediments. Whole metagenome and amplicon sequencing, metabolomics, geochemistry and thermodynamic modelling were deployed towards the goals of this investigation. The use of the NextSeq sequencing platform allowed for large amounts of data to be generated which led to a remarkable reconstruction of 82 MAGs with >50% completeness. This is probably one of the most extensive metagenomic studies of deep water environments and the supplementary metabolomic and geochemical data make it a rather interesting manuscript.

General Comments:

- The article is well-written, and with a proper use of the English language.
- The article however feels a bit descriptive and the justification for the study seems a little bit vague. Please ensure that the specific questions/hypothesis are made clearer in the beginning of the manuscript.
- It appears that the three sediment samples mentioned here are a subset of a greater sampling effort published previously by the same group. What was the rationale of using these three sediment samples amongst the available samples? Were they chosen based on their similarities in terms of microbial

community composition (and hence used as pseudoreplicates) or where they meant to be used as contrasting cases of deep-water microbial communities? Either way, the authors did not elaborate enough on the comparison of these three samples. It felt that the same conclusions would have reached with any of these three samples and there were no significant differences in their predicted microbial processes despite the quite different geochemical signatures. If there were differences (as some of the supplementary material might suggest), the authors could have done a better job at better articulating them in the manuscript.

- Handling and presenting large datasets such as metagenomic and metabolomic data can be challenging. Listing the metabolomic data in the supplementary material does not make justice to them. The authors might want to consider using reference pathway maps (like those in KEGG or MetaCyc) to overlay both the metagenomic and metabolomic data. That would facilitate an easier visual depiction of the biochemical pathways that are found in the metagenomic datasets and which ones are in fact active based on the metabolites identified in these environments. This will be particularly informative on hydrocarbon degradation and carbon-cycling pathways since they are the main focus of this work.

- Several other studies on cold seep sediments including those in the Gulf of Mexico identify a large population of ANaerobic MEthanotrophs (ANME lineages). It is surprising that no ANME's were detected in these seepage locations especially at location E29 which contained a large concentration of methane. Could the authors propose any plausible reasons that explain the absence of ANMEs from these sediments?

Specific Comments:

L45: Given the provided references the "subsurface microbes" should be changed to "subseafloor microbes".

L49: Change "subseafloor" to "subsurface"

L95: Sediment E29 can indeed be predominantly biogenic ("microbial"). What is the evidence for the other two? Also, how do the authors explain the biogenic isotopic signature of methane and the presence of more complex hydrocarbons in the same samples? Is it coincidental?

L114: What was the rationale of performing amplicon sequencing in addition to the whole metagenome sequencing? How does the derived taxonomy from the 16S rRNA amplicon data compare with the one derived from the 16S rRNA data recovered from the metagenomes?

L136-L145: The information on candidate phylum TA06 seems disproportionate the other identified phyla. Consider revising its length or justify the special focus on that one.

L171: if this indeed the aim of the study consider moving it closer to the beginning of the manuscript.

L245: Change to "The potential for H₂ metabolism was also found ..."

L256: Change to "... corresponding genes frequently co-occur ..."

L264: Could the authors speculate as to why only two of the dominant MAGs in Figure 3 contained the *dsrAB* gene?

L271: According to Figure 3, one third of the *rdhA* containing MAGs did not use the W-L or TCA cycle. Please explain or revise the sentence.

L363: None of these 12 MAGs appear to contain a *dsrAB* gene. Please explain the claim that they can be sulfate reducers or revise the sentence.

L382: what is the evidence that leads to the estimation of “turnover rates”? Please explain or revise the sentence.

L403: Where there any known hydrocarbon seeps nearby? If so, how far from the seeps were the samples taken? Where any replicate samples taken from the same location? Did the authors examine any deeper samples from these cores? If so, did their microbial community differ from those near the surface?

L452: change “v4-8” to “v4-5”

Table 1: Add methane concentration to the table if available.

Figure 3: This figure might benefit from adding the MAG % completeness next to the name.

Figure 6: This figure might benefit from changing the arrow thickness to indicate predominant processes and also by positioning the dominant lineages to the corresponding arrows.

B) Validity:

No flaws were detected.

C) Originality and significance:

The results are interesting, but the conclusions mostly confirm previous studies (Arndt et al., 2013; Beulig et al., 2018). The hydrocarbon degradation element could have been better presented.

D) Data & methodology:

The data are satisfactorily detailed and transparent

E) Appropriate use of statistics and treatment of uncertainties:

Comparison between samples could benefit from an associated similarity score

F) Conclusions:

Conclusions seem robust enough although they could benefit from better data integration and visualization.

G) Suggested improvements:

Suggested improvements are mentioned in the sections above.

H) References:

All references are appropriate

I) Clarity and context:

For the most part it is clear and accessible. The authors could have done a better job at highlighting the novelty and aim of the study and how that differs from previously published studies.

Response to Referees

Reviewer #1 (Remarks to the Author):

In this work, the authors conducted massive metagenome sequencing on three sediment samples from hydrocarbon-impacted deep sea sediments in the Gulf of Mexico. The upper 20 cm sediment was investigated. In total, 82 genomes were binned from the metagenome data, which is an excellent outcome considering the high diversity of marine sediments. Identification of hydrocarbon-activating enzymes was pursued not only by following automated annotation pipelines, but also by direct search using known enzymes, which adds to the robustness of the analysis. The work was complemented by sediment geochemistry, diversity based on amplicon sequencing, and metabolomics. Reconstruction of metabolic potential of those sediments from the data resulted in a complex metabolic network, including oxidation of detrital biomass, hydrocarbons, aromatic compounds, acetate and hydrogen.

Although the data generated is robust and adds to our knowledge of the metabolic potential of deep sea sediments, the interpretation suffers from inconsistencies, over-interpretation or unsupported conclusions that need to be carefully considered (see below).

Response: We appreciate this careful review and feedback. We have revised the manuscript in line with the comments (see responses below).

Although the work is introduced as focused on anaerobic hydrocarbon oxidation (Title), this is misleading since the rest of the manuscript (from Abstract on) is rather focused on the whole metabolic potential of these sediments. This does not mean the work is less valuable, but the focus should be clearly stated. I recommend changing the title to better reflect the contents, and guide the expectations of the reader.

Response: We agree with this summary and that the original title did not fully reflect what we have presented. Therefore, we revised the title to: “Metabolic potential of uncultured bacteria and archaea associated with petroleum seepage in deep-sea sediments”.

The study does not appear to have uncovered a very high potential for anaerobic hydrocarbon oxidation. The only proteins directly related to anaerobic hydrocarbon activation are the divergent AssA, for alkane activation.

Response: We revised the paragraphs related to anaerobic hydrocarbon degradation to highlight our findings on both hydrocarbon addition to fumarate and hydroxylation with water by molybdenum cofactor-containing enzymes. In addition, as elaborated below, we have performed a more in-depth analysis of the metabolomics analysis, which now provides very convincing support for the occurrence of the hydrocarbon addition to fumarate activation pathway and benzoyl-CoA degradation pathway.

L159-170: “We obtained evidence that two of the four known pathways for oxygen-independent C-H activation ²⁵⁻²⁸ were present: hydrocarbon addition to fumarate by glycy radical enzymes ²⁸ and hydroxylation with water by molybdenum cofactor-containing enzymes ²⁵. Glycy radical enzymes proposed to mediate hydrocarbon addition to fumarate were found in 13 MAGs ... Based on quality-filtered reads, canonical AssA (n-alkane succinate synthase) and BssA (benzyl succinate synthase) enzymes are also encoded at these sites and were most abundant in Site E29 (Supplementary Table 7).”

L172-175: “For the hydroxylation pathway, a *Dehalococcoidia* MAG in Site E29 encoded proteins sharing over 40% sequence identities to the catalytic subunits of *p*-cymene dehydrogenase (Cmd) and alkane C₂-methylene hydroxylase (Ahy) (Figure 3 and Supplementary Figure 4) ³².”

This is supported to some extent by metabolomics, where some alkyl-succinates have been found (although the metabolomics data are not very visible in the manuscript).

Response: We thank the reviewer for this comment, which, together with similar suggestions from the other two reviewers, motivated us to conduct a more systematic metabolomics analysis using five replicates from each of the three sediments. The resulting new data show strong evidence for hydrocarbon degradation despite the low abundance of canonical hydrocarbon-degrading genes. This is now highlighted in a new metabolomics heat map (Figure 4) and Results.

L170-172: “In agreement with this, metabolomics analysis detected six succinic acid conjugates involved in hydrocarbon activation, including conjugates of xylene, toluene, cyclohexane, and isopropane (Figure 4).”

L180-185: “Through metabolomic analysis, we detected multiple intermediates involved in both the production and degradation of benzoyl-CoA, a universal intermediate formed during the degradation of aromatic compounds³³ (Figure 4). Various compounds that can be activated to form benzoyl-CoA were detected, including benzoate, benzylsuccinate, 4-hydroxybenzoate, phenylacetate, acetophenone, and phenol. The downstream metabolite glutarate was also highly abundant (Figure 4).”

No direct evidence for the anaerobic oxidation of aromatic hydrocarbons was provided. The enzymes for anaerobic benzoate oxidation cannot be used as indicators of hydrocarbon oxidation: it is true that benzoate is an intermediate in the anaerobic oxidation of many aromatic hydrocarbons, but it can also be formed from other sources. Bss/Nms would provide direct evidence for aromatic hydrocarbon oxidation, but encoding genes were not found.

Response: We agree that the gene evidence is not strong to support the occurrence of anaerobic oxidation of aromatic hydrocarbons. However, *bssA* genes were indeed found in quality-filtered reads. More importantly, the current metabolomics now provides strong evidence for anaerobic oxidation of aromatic hydrocarbons, based on the identification of metabolites, *e.g.* benzylsuccinate for toluene degradation. See Figure 4 and Results.

L162-170: “Glycyl-radical enzymes proposed to mediate hydrocarbon addition to fumarate were found in 13 MAGs ...Based on quality-filtered reads, canonical AssA (n-alkane succinate synthase) and BssA (benzyl succinate synthase) enzymes are also encoded at these sites and were most abundant in Site E29 (Supplementary Table 7).”

L170-172: “In agreement with this, metabolomics analysis detected six succinic acid conjugates involved in hydrocarbon activation, including conjugates of xylene, toluene, cyclohexane, and isopropane (Figure 4).”

Considering the above, some statements regarding the potential for anaerobic hydrocarbon oxidation should be toned down (for ex. L205, 336). Also, the oxidation of aromatic compounds should be in a way decoupled from that of aromatic hydrocarbons, or make clear that the finding of Bcr for example is not proof for hydrocarbon oxidation.

Response: Although specific metabolites associated with anaerobic hydrocarbon oxidation were identified, we agree that benzoate could also originate from other sources besides anaerobic oxidation of aromatic hydrocarbons. We therefore now refer to “aromatic compounds” in the text, instead of “aromatic hydrocarbons” in several places. We now clearly highlight the differences between aromatic compounds and aromatic hydrocarbons.

L179-180: “Our results also provide evidence that aromatic compounds can be anaerobically degraded *via* channeling into the central benzoyl-CoA degradation pathway.”

L197-198: “However, sources of benzoate other than via anaerobic degradation of the above compounds, cannot be ruled out based on current data.”

L277-278: “Instead, we provide theoretical evidence that anaerobic degradation of aliphatic and aromatic compounds is feasible in this environment by...”

‘Fermentation’ is used in a wrong way when applied to alkane (hexadecane) oxidation (Section Thermodynamic modelling of hydrocarbon degradation). Alkanes are reduced compounds which cannot be fermented. They are indeed partly oxidized to acetate and hydrogen (reduction of protons) in syntrophic reactions (Zengler, K., et al., Methane formation from long-chain alkanes by anaerobic microorganisms. Nature 401, 266-269, 1999): the correct term would be partial or incomplete oxidation, or syntrophic oxidation. Benzoate can be fermented, but the products are acetate and cyclohex-1-ene carboxylate (e.g. Elshahed, M. S. & McInerney, M. J. Benzoate Fermentation by the Anaerobic Bacterium Syntrophus aciditrophicus in the Absence of Hydrogen-Using Microorganisms. Appl. Environ. Microbiol. 67, 5520-5525, 2001). Syntrophic benzoate oxidation would also yield acetate and hydrogen (same reference as above).

Response: We thank the reviewer for these insightful comments and agree that the term “fermentation” was improperly used. This has been changed to “incomplete oxidation” as suggested.

L282: “...likely that anaerobic degradation occurs through an incomplete oxidation pathway...”

L287-290: “... anaerobic biodegradation for three plausible scenarios (Table 2): (i) incomplete oxidation with production of hydrogen and acetate, (ii) acetogenic oxidation with production of acetate alone, and (iii) complete oxidation coupled with sulfate reduction...”

In view of the above, what is the basis for the proposed reactions (Fig. 5) yielding only acetate and no H₂? What would be the electron acceptor in that case?

Response: In addition to metagenomic evidence for Wood-Ljungdahl pathway genes, our inclusion of acetogenic oxidation of hexadecane and benzoate was inspired by other recent studies (References #37, #40, and #49). However, the reviewer is correct that we provide no direct evidence of these reactions occurring in these environments. In light of this, we have toned down the language describing this scenario at L285-290, L293-299 and in the thermodynamics figure (now Figure 5) as well as Table 2.

L285-290: “Additionally, several recent studies indicate that some aliphatic and aromatic compounds can be incompletely oxidized to acetate *via* the Wood-Ljungdahl pathway^{37, 40, 49}. Therefore, we compared the thermodynamic constraints on anaerobic biodegradation for three plausible scenarios (Table 2): (i) incomplete oxidation with production of hydrogen and acetate, (ii) acetogenic oxidation with production of acetate alone, and (iii) complete oxidation coupled with sulfate reduction.”

L293-299: “Under deep-sea conditions, without taking the *in situ* concentrations of hydrogen and acetate into consideration, the calculations show that sulfate-dependent complete oxidation of hexadecane and benzoate, as well as acetogenic oxidation of benzoate, would be energetically favorable. The three other reactions would be endergonic under most conditions, but based on the measured concentrations for both acetate and H₂ in these sediments being low (Supplementary Note 2) these reactions could also take place in theory (Table 2 and Figure 5).”

Also, the author's conclusion that hydrocarbons are mostly incompletely oxidized (Line 296 and fwd.) is surprising, and should be reconsidered: almost all MAGs recovered with AssA-like proteins contain (according to Fig. 3) either WL pathway or TCA cycle enzymes (or even both). Both pathways can operate in reverse, and can be used for complete oxidation of acetate to CO₂. Also, it is stated in the section ahead (L262 and fwd.) that genes for sulfate reduction and dehalorespiration have been found across the metagenome reads. Absence of specific pathways for the reduction of a terminal electron acceptor in AssA-containing bins may be due to binning or incomplete genomes, for example.

Response: We agree with this comment. Due to the binning process or incompleteness of genomes, some genes might be missing from MAGs such as *dsrAB*. Therefore, we cannot exclude the possibility of complete oxidation of alkane and aromatic compounds coupled with reduction of *e.g.* sulfate. We have incorporated this scenario into the revised manuscript. On the other hand, since most of these lineages do not contain known SRB and other studies have suggested these groups are most likely acetogens/fermenters, we have retained in our analysis the scenario of incomplete oxidation as well. As a result, the revised manuscript presents a more balanced range of possible metabolisms summarized by three kinds of scenarios, shown in Table 2. See L282-290, Table 2 and Figure 5.

L282-290: “However, due to incompleteness of the reconstructed genomes, we cannot exclude the possibility that complete oxidation of aliphatic and aromatic compounds to CO₂ occurs through, for example, coupling with sulfate reduction (Figure 3). Additionally, several recent studies indicate that some aliphatic and aromatic compounds can be incompletely oxidized to acetate *via* the Wood-Ljungdahl pathway^{37, 40, 49}. Therefore, we compared the thermodynamic constraints on anaerobic biodegradation for three plausible scenarios (Table 2): (i) incomplete oxidation with production of hydrogen and acetate, (ii) acetogenic oxidation with production of acetate alone, and (iii) complete oxidation coupled with sulfate reduction.”

Conclusion at L309-315 is wrong: how could an energy-demanding CO₂ fixation pathway (WL, CO₂ to acetate) be energetically more favorable than an energy-conserving one (incomplete alkane oxidation), no matter how less energetically favorable is the latter, when both end up in the same acetate pool? See energetics of syntrophic hexadecane oxidation in methanogenic

cultures (Zengler, K., et al., Methane formation from long-chain alkanes by anaerobic microorganisms. Nature 401, 266-269, 1999).

Response: Thank you. This part has been removed from the revised manuscript.

L355-357: this is an understatement, in view of the amount of literature published in the past decade on anaerobic hydrocarbon degradation; and there are plenty of studies with marine cultures (AOM literature, nonmethane gaseous alkane degraders (e.g. Kniermeyer, O., et al., Anaerobic oxidation of short-chain hydrocarbons by marine sulphate-reducing bacteria. Nature 449, 898-U810, 2007), strains AK-01 and Hxd3 (Univ. of Oklahoma publications), etc.

L357-361: there is no basis for these statements, and they should be removed. 'Mechanisms' refers to the biochemistry of hydrocarbon activation, and the present work does not provide hints to novel mechanisms.

Response: We appreciate these useful comments. We removed both inappropriate statements.

Other suggestions for improvement

Overall, the manuscript may benefit from shortening and better streamlining the main ideas. The relevance of some sections is not right away clear, like the section at L136-145, dedicated to candidate phylum TA06.

Response: We shortened the whole manuscript and focused on the main ideas. We agree that the section on TA06 is not of primary relevance and therefore moved it to supplementary materials (Supplementary Note 1).

L100: methylsilyl esters are not metabolites. They are formed by sample derivatization upon extraction.

Response: We have removed this statement from the text and provided new data on the presence of alkyl-/arylalkylsuccinates (Figure 4 and L170-172).

L170-172: “In agreement with this, metabolomics analysis detected six succinic acid conjugates involved in hydrocarbon activation, including conjugates of xylene, toluene, cyclohexane, and isopropane (Figure 4).”

L183 and elsewhere: Gly-radical enzymes of hydrocarbon activation do not catalyze ‘fumarate addition’ although this is a phrasing used often. They catalyze the cleavage of the hydrocarbon C-H bond, which then adds to fumarate (which is a co-substrate). The correct reference is ‘hydrocarbon addition to fumarate’

Response: We changed “fumarate addition” to “hydrocarbon addition to fumarate”. See L160 and as well as in other parts of the manuscript.

L173-174: C-H bonds are not mechanistically sophisticated: they are simple, stable, apolar bonds.

Response: We changed “sophisticated” to “stable”. See L159.

L235: there are no such things as heterotrophic compounds

Response: We corrected this. L209-210: “...using a range of inorganic and organic substrates”.

L314/15: here benzoate and related compounds are listed as hydrocarbons.

Response: This has been corrected; throughout the manuscript we now refer to “aromatic compounds” to avoid this error.

Reviewer #2 (Remarks to the Author):

Dong and coauthors report 82 new MAGs from three hydrocarbon-rich surface sediments. They discuss new potential roles for uncultured organisms in hydrocarbon degradation, and the importance of hydrogen and acetate as intermediates. In general, this work seems straightforward and useful. I just have a few points of interest.

The abstract and the methods refer to having performed metabolomics. However, it appears that only benzoate and succinate are listed in Table 1. I don’t think it’s a problem if many metabolites

were measured, but the authors only discuss these two. However, it is confusing to expect an array of metabolites, and then only really have two that are buried in Table 1. I say buried because they seem like they were part of a separate geochemical assay designed to measure each compound, as is true for all the other data in Table 1. I think the authors should discuss how many other metabolites were identified and why they are not analyzed. Also, the succinate should be discussed in the text somewhere.

Response: We thank the reviewer for this comment, which together with comments from the other two reviewers inspired us to re-analyse our samples with larger injection volumes and greater replication. Over 50 compounds from a diverse transect of metabolic pathways were observed. Selected metabolites are now removed from Table 1, and instead all metabolites are presented together in a new heat map. See Figure 4, L149-153, L170-172 and L181-186.

L150-154: “Metabolomics data supported these genomic predictions and showed a surprising degree of consistency between the geographically distinct sampling sites (Figure 4). Over 50 metabolites from eight pathways were detected in all the samples, including carbohydrate metabolism (*e.g.* glucose), amino acid metabolism (*e.g.* glutamate), and beta oxidation (*e.g.* 10-hydroxydecanoate).”

L170-172: “In agreement with this, metabolomics analysis detected six succinic acid conjugates involved in hydrocarbon activation, including conjugates of xylene, toluene, cyclohexane, and isopropane (Figure 4).”

L180-185: “Through metabolomic analysis, we detected multiple intermediates involved in both the production and degradation of benzoyl-CoA, a universal intermediate formed during the degradation of aromatic compounds³³ (Figure 4). Various compounds that can be activated to form benzoyl-CoA were detected, including benzoate, benzylsuccinate, 4-hydroxybenzoate, phenylacetate, acetophenone, and phenol. The downstream metabolite glutarate was also highly abundant (Figure 4).”

I was not sure how sulfate could go so high above 28 mM. I'm not aware of any methods to concentrate sulfate in marine sediments – is it possible that the calibration curves were inaccurate?

Response: We suspected that such unusually high concentrations could have resulted from inconsistencies in the sample volume being injected into the column by the autosampler of our ion chromatography (IC) system. A reproducibility test by multiple injections from one sample confirmed our suspicion and we therefore replaced the faulty autosampler needle. After being satisfied with the accuracy of sample injection by the new sampler and the precision of the calibration curve ($R^2 = 0.99$), we re-analyzed the sulfate and chloride concentrations of freshly extracted pore water from the three sediments. Triplicate injections were made from each sample vial into the column. As expected, the sulfate concentrations were all below 28 mM and the values of the replicate injections from each sample were highly reproducible with standard deviations being in the range of 0.01-0.04 mM. The chloride concentrations were also slightly lower than what was reported in the original manuscript. We further confirmed these values by analyzing the same pore water samples using a similar IC system in a different lab. Based on these analyses, we have revised the sulfate concentrations, which are now presented in a Supplementary Note, and referenced at Line 242.

Supplementary Note 2: “We performed a geochemical analysis of sediment porewater extracts. High concentrations of sulfate (Site E26: 16.55 mM; Site E29: 27.23 mM; Site E44: 25.63 mM) were detected at each of the three sites, consistent with sulfate being 28 mM in seawater and diffusing into sediments where it is consumed by sulfate reduction under anoxic conditions...”

L242: “In agreement with porewater sulfate concentrations (16-27 mM; see Supplementary Note 2)...”

I find it very surprising that no methanogens or mcrA were found, when the $d^{13}C$ of the methane clearly indicates methanogenesis. I agree with the authors that a likely explanation is that methanogens could be deeper than the samples. However, the samples were in the upper 20 cm, which should be sufficiently deep to avoid oxygen. The lack of aerobic genes supports this. Do the authors know of other instances of petroleum-rich sites where methanogens were absent?

Response: We are not aware of any petroleum-rich sites where methanogens were absent. However, as the reviewer mentioned, these sediments used for genomic analysis are anaerobic, with sulfate levels suggesting that they are in the sulfate reduction zone but above the sulfate-

methane transition zone (SMTZ). In such environments, methanogens should be outcompeted by SRB. Therefore, most likely methanogenesis does not occur in this sampling horizon used for genome analysis. Similar phenomena have also been reported in other studies where researchers had limited observations of methanogens in contrast with the high geochemistry signals for biological methane production (Carr et al. 2017). We provided a better explanation of this, and now include this example from the literature (Reference #48), at L264-270.

L264-270: “Overall, the collectively weak *mcrA* signal in the metagenomes suggests that the high levels of biogenic methane detected by geochemical analysis (Table 1) is due to methanogenesis in deeper sediment layers. Similar phenomena have been observed in other sites, where *mcrA* genes are in low abundance despite clear geochemical evidence for biogenic methane⁴⁸. Sequencing additional sediment depths at greater resolution would likely result in detection of *mcrA*-related methanogens and ANME lineages.”

They might consider mentioning the possibility that some of these hydrocarbon-degrading pathways that are so abundant produce methane independently of mcrA.

Response: There are a few methane production pathways known that don't involve methanogenic archaea or *mcrA*, including methylphosphonate decomposition, algal production, and Fe-nitrogenase activity. However, we don't think these would be particularly relevant in the ecosystem we studied here. To the best of our knowledge, we are not aware of any hydrocarbon degradation pathways that directly produce methane independently of *mcrA*.

Figure 6 seems really uninformative. All the processes mentioned are things that are known to occur in hydrocarbon-rich marine sediments. It seems that the novel addition of this paper is that uncultured groups appear to participate in this process. Perhaps this information could be included in this figure.

Response: We agree that this figure did not add very much to the manuscript and therefore removed it. The key finding of the importance of uncultured groups is emphasized at the beginning of the Introduction and the beginning of the Discussion so that this message gets across without requiring that figure.

The conclusion that deep-sea sediment communities are electron-donor driven and coastal sediments are electron acceptor driven (lines 333-335) seems like a strange distinction. Why would this be the case? Also, it is based on the fact that terminal respiring organisms appear to be in low abundance in these samples, but studies such as Leloup et al. 2009, EMI, suggest that this is also the case in coastal sediments.

Response: We revised the sentences according to the reviewer's comment to emphasize more generally the importance of substrates in deep-sea sediments without drawing the unnecessary and potentially inaccurate comparison to shallow coastal sediments.

L317-318: "Therefore, microbial communities in the deep seabed are likely shaped more by the capacity to utilize available electron donors than by the availability of oxidants."

In summary, I think that the novel findings in this paper are the potential ability of deeply-branching uncultured clades to degrade hydrocarbons at deep-sea seeps. However, these findings are buried among much more general statements. For instance, the title says nothing about this. We have long known that anaerobic hydrocarbon degradation occurs in deep-sea sediments. I suggest that the authors bring their novel discoveries to the forefront, starting with the title.

Response: We agree with the reviewer's summary of the novel findings. We have changed the Title, Introduction and Discussion to focus on the potential capabilities of uncultured bacteria and archaea.

Title: "Metabolic potential of uncultured bacteria and archaea associated with petroleum seepage in deep-sea sediments".

L70-72: "In this study, we used culture-independent approaches to study the role of microbial communities in the degradation of organic matter, including both detrital biomass and petroleum hydrocarbons."

L341-346: "The finding that Bathyarchaeota and other archaeal phyla are potentially capable of anaerobic degradation of aliphatic or aromatic compounds extends the potential substrate

spectrum for archaea. More broadly, building on recent findings²⁶, this work emphasize that non-methane aliphatic and aromatic compounds could significantly contribute to carbon and energy budgets in these deep-sea settings.”

Reviewer #3 (Remarks to the Author):

A) Key results:

The authors used multiple sources of information to investigate microbial processes and carbon cycling in deep seawater sediments. Whole metagenome and amplicon sequencing, metabolomics, geochemistry and thermodynamic modelling were deployed towards the goals of this investigation. The use of the NextSeq sequencing platform allowed for large amounts of data to be generated which led to a remarkable reconstruction of 82 MAGs with >50% completeness. This is probably one of the most extensive metagenomic studies of deep water environments and the supplementary metabolomic and geochemical data make it a rather interesting manuscript.

General Comments:

- *The article is well-written, and with a proper use of the English language.*
- *The article however feels a bit descriptive and the justification for the study seems a little bit vague. Please ensure that the specific questions/hypothesis are made clearer in the beginning of the manuscript.*

Response: We have revised the Introduction to make the study questions clearer. See L70-72: “In this study, we used culture-independent approaches to study the role of microbial communities in the degradation of organic matter, including both detrital biomass and petroleum hydrocarbons.”

- *It appears that the three sediment samples mentioned here are a subset of a greater sampling effort published previously by the same group. What was the rationale of using these three sediment samples amongst the available samples? Were they chosen based on their similarities in terms of microbial community composition (and hence used as pseudoreplicates) or where they meant to be used as contrasting cases of deep-water microbial communities?*

Response: Samples used in this study were a subset of a greater sampling effort. The samples were chosen based on having different associated hydrocarbon geochemistry profiles. We have added detailed explanations on why we selected these three samples in the Introduction and Methods.

Introduction (L72-74): “We performed metagenomic, geochemical and metabolomic analyses of deep seabed sediments (water depth ~3 km). Samples were chosen from three sites exhibiting evidence of different levels of migrated thermogenic hydrocarbons.”

Methods (L379-397): the whole section of “Sample selection based on geochemical characterization” starting from “The three marine sediment samples used in this study were chosen from among several sites sampled as part of a piston coring seafloor survey in the Eastern Gulf of Mexico, as described previously...”

Either way, the authors did not elaborate enough on the comparison of these three samples. It felt that the same conclusions would have reached with any of these three samples and there were no significant differences in their predicted microbial processes despite the quite different geochemical signatures. If there were differences (as some of the supplementary material might suggest), the authors could have done a better job at better articulating them in the manuscript.

Response: We indeed found some differences in terms of microbial compositions and abundance of some functional metabolic genes. However, in terms of metabolic potential of reconstructed MAGs, as the reviewer points out, we found no significant differences in predicted microbial functions. This suggested functional redundancy similar to a recent study in Guaymas Basin hydrothermal sediments (Reference #50). We have added and highlighted these details in the Results and Discussion.

L113-118: “While the three sites share a broadly similar community composition, notable differences were *Ca.* Bathyarchaeota and Proteobacteria being in higher relative abundance at the sites associated with more hydrocarbons (E29 and E26; Table 1), whereas the inverse is true for Actinobacteria, the Patescibacteria group, and *Ca.* Aerophobetes that are all present in higher relative abundance at Site E44 where associated hydrocarbon levels are lower.”

L168-170: “Based on quality-filtered reads, canonical AssA (n-alkane succinate synthase) and BssA (benzyl succinate synthase) enzymes are also encoded at these sites and were most abundant in Site E29 (Supplementary Table 7).”

L318-324: “In line with the different geochemical profiles at the three sites (Table 1), some differences in the composition of microbial communities and the abundance of key metabolic genes were observed (Figure 1 and Supplementary Table 7). However, metabolic capabilities such as fermentation, acetogenesis, and hydrogen metabolism were conserved across diverse phyla in each site (Figure 3). This suggests some functional redundancy in these microbial communities, similar to that recently inferred in a study of Guaymas Basin hydrothermal sediments⁵⁰.”

• *Handling and presenting large datasets such as metagenomic and metabolomic data can be challenging. Listing the metabolomic data in the supplementary material does not make justice to them. The authors might want to consider using reference pathway maps (like those in KEGG or MetaCyc) to overlay both the metagenomic and metabolomic data. That would facilitate an easier visual depiction of the biochemical pathways that are found in the metagenomic datasets and which ones are in fact active based on the metabolites identified in these environments. This will be particularly informative on hydrocarbon degradation and carbon-cycling pathways since they are the main focus of this work.*

Response: We thank the reviewer for their understanding in dealing with metabolomics data and encouraging showcasing the metabolomics more prominently. As mentioned already in response to both of the other two reviewers, we have incorporated a more extensive metabolomics dataset into the revised manuscript, and highlight relevant metabolic pathways in Figure 4. Our new figure lists the observed hydrocarbon degradation metabolites that have been previously described and mapped by Gieg and Toth (2017). We have made more explicit references to these pathways to aid readers in the understanding of these metabolic networks. As the reviewer suggested, the manuscript is more informative now with key messages more strongly supported. See Methods, Figure 4, Results.

L150-154: “Metabolomics data supported these genomic predictions and showed a surprising degree of consistency between the geographically distinct sampling sites (Figure 4). Over 50 metabolites from eight pathways were detected in all the samples, including carbohydrate metabolism (*e.g.* glucose), amino acid metabolism (*e.g.* glutamate), and beta oxidation (*e.g.* 10-hydroxydecanoate).”

L170-172: “In agreement with this, metabolomics analysis detected six succinic acid conjugates involved in hydrocarbon activation, including conjugates of xylene, toluene, cyclohexane, and isopropane (Figure 4).”

L180-185: “Through metabolomic analysis, we detected multiple intermediates involved in both the production and degradation of benzoyl-CoA, a universal intermediate formed during the degradation of aromatic compounds³³ (Figure 4). Various compounds that can be activated to form benzoyl-CoA were detected, including benzoate, benzylsuccinate, 4-hydroxybenzoate, phenylacetate, acetophenone, and phenol. The downstream metabolite glutarate was also highly abundant (Figure 4).”L408-430: the whole section of “Metabolomic analysis” starting from “For the analysis of metabolites...”

• *Several other studies on cold seep sediments including those in the Gulf of Mexico identify a large population of ANaerobic MEthanotrophs (ANME lineages). It is surprising that no ANME’s were detected in these seepage locations especially at location E29 which contained a large concentration of methane. Could the authors propose any plausible reasons that explain the absence of ANMEs from these sediments?*

Response: As indicated in response to Reviewer #2 this apparent discrepancy is likely related to the sampling depth being above the sulfate-methane transition zone. Sampling depth is now explained more clearly in the Methods section.

Methods (L379-397): the whole section of “Sample selection based on geochemical characterization” starting from “The three marine sediment samples used in this study were chosen from among several sites sampled as part of a piston coring seafloor survey in the Eastern Gulf of Mexico, as described previously...”

Specific Comments:

L45: Given the provided references the “subsurface microbes” should be changed to “subseafloor microbes”.

Response: changed as suggested. See L42.

L49: Change “subseafloor” to “subsurface”

Response: changed as suggested. See L47.

L95: Sediment E29 can indeed be predominantly biogenic (“microbial”). What is the evidence for the other two?

Response: Although we did not directly measure methane concentrations, they are approximately equal to the values obtained by subtracting C₂₊ Alk (Sum of alkane gases less methane) from ΣAlk Gas (Total Alkane Gases). Based on the data shown in Table 1, the methane levels in the other two samples are very low, consistent with our inability to measure stable isotope ratios for them. This information has now been added in the footnote of Table 1.

Also, how do the authors explain the biogenic isotopic signature of methane and the presence of more complex hydrocarbons in the same samples? Is it coincidental?

Response: Yes, this could happen. Similar findings with co-existence of biogenic methane and complex hydrocarbons have been also reported in other studies. For example, in one hydrocarbon seep in the Mississippi Canyon 118 (MC118) in the Gulf of Mexico, methane was also shown to be both thermogenic and biogenic (Lapham et al., 2008). We now make reference to this study in the Introduction.

L96-97: “Similar co-occurrence of biogenic methane and complex hydrocarbons have been reported in a nearby seep in the Mississippi Canyon in the Gulf of Mexico²².”

L114: What was the rationale of performing amplicon sequencing in addition to the whole metagenome sequencing?

Response: The amplicon sequencing was completed first and provided fast and inexpensive assessments of bacterial and archaeal communities. More importantly, our study uses amplicon sequencing to explain the quantitative insights into microbial diversity as described in L105-118. We added this information to the Methods section (L443-444): “To provide a high-resolution microbial community profile, as well as quantitative insights into microbial community diversity...”.

How does the derived taxonomy from the 16S rRNA amplicon data compare with the one derived from the 16S rRNA data recovered from the metagenomes?

Response: The values are not exactly the same in terms of relative abundance, probably due to *e.g.* the preferential amplification of certain taxa in 16S rRNA gene amplicon sequencing. However, the two methods are in broad agreement with each other at least for these abundant phyla. We added this in L108-113: “In accordance with amplicon sequencing results, taxonomic profiling of metagenomes using small subunit ribosomal RNA (SSU rRNA) marker genes demonstrated that the most abundant phyla in the metagenomes were, in decreasing order, Chloroflexi (mostly classes *Dehalococcoidia* and *Anaerolineae*), *Candidatus* Atribacteria, Proteobacteria (mostly class *Deltaproteobacteria*), and *Candidatus* Bathyarchaeota (Supplementary Table 3 and Figure 1a).”.

L136-L145: The information on candidate phylum TA06 seems disproportionate the other identified phyla. Consider revising its length or justify the special focus on that one.

Response: We agree with this suggestion and have moved this paragraph into the supplementary materials as Supplementary Note 1.

L171: if this indeed the aim of the study consider moving it closer to the beginning of the manuscript.

Response: **Thank you for this important suggestion.** We revised these sentences and have now clearly stated our goal in the Introduction.

L70-72: “In this study, we used culture-independent approaches to study the role of microbial communities in the degradation of organic matter, including both detrital biomass and petroleum hydrocarbons.”

L157: “To identify the potential for microbial degradation of hydrocarbons, we focused...”.

L245: Change to “The potential for H₂ metabolism was also found ...”

Response: Changed as suggested. L223.

L256: Change to “... corresponding genes frequently co-occur ...”

Response: Changed as suggested. L234.

L264: Could the authors speculate as to why only two of the dominant MAGs in Figure 3 contained the dsrAB gene?

Response: This was also pointed out by Reviewer #1. The result that only two MAGs contained *dsrAB* genes might be due to binning or incomplete genomes. These explanations have been added at L244-246: “...however, probably due to incompleteness of genomes or insufficient binning, these genes were identified only in two MAGs affiliated with *Deltaproteobacteria* and *Dehalococcoidia* (Supplementary Table 6)”.

L271: According to Figure 3, one third of the rdhA containing MAGs did not use the W-L or TCA cycle. Please explain or revise the sentence.

Response: We revised the sentence. L250: “At least two thirds of the MAGs corresponding to putative sulfate reducers and dehalorespirers...”.

L363: None of these 12 MAGs appear to contain a dsrAB gene. Please explain the claim that they can be sulfate reducers or revise the sentence.

Response: We revised the sentence. See L347-348: “Genomic analyses of the 12 MAGs harboring genes for central benzoyl-CoA pathway indicate they have an acetogenic lifestyle.”

L382: what is the evidence that leads to the estimation of “turnover rates”? Please explain or revise the sentence.

Response: Yes, this statement was confusing. We revised the sentence to make it clear. See L365-367: “Despite the presence of putative acetogens and hydrogenogens, acetate and hydrogen were below the limits of detection in sediment porewater (Supplementary Note 2), indicating both compounds are rapidly turned over by other community members.”

L403: Where there any known hydrocarbon seeps nearby? If so, how far from the seeps were the samples taken? Where any replicate samples taken from the same location? Did the authors examine any deeper samples from these cores? If so, did their microbial community differ from those near the surface?

Response: All three locations in this study are located in the deep Abyssal Plain province within the Gulf of Mexico. The seafloor in this region has not been explored well enough compared to the north-western (*e.g.* Mississippi Canyon, Green Canyon) and south-western (*e.g.* Chapopote and Campeche Knolls) shelves of the Gulf where most seep discoveries have been to date. We therefore are not aware of any well-known hydrocarbon seeps in the immediate vicinity of these locations. Since the samples were collected as part of a larger surface geochemical survey (>100 locations), only one core per location was obtained in order to cover a large area. Consequently, collection of ecological replicates and sample preservation for microbiology from the deeper sections were not possible. We added one figure into supplementary information to make the locations more visible to readers (Supplementary Figure 1 and L87). Microbial communities in deeper samples were not examined.

L452: change “v4-8” to “v4-5”

Response: Changed as suggested. L448.

Table 1: Add methane concentration to the table if available.

Response: We did not directly measure methane concentrations. However, they are approximately equal to the values obtained by subtracting C₂₊ Alk (Sum of alkane gases less methane) from ΣAlk Gas (Total Alkane Gases). See footnotes of Table 1.

Figure 3: This figure might benefit from adding the MAG % completeness next to the name.

Response: This is a great suggestion, and the completeness % has now been added. Now revised Figure 3.

Figure 6: This figure might benefit from changing the arrow thickness to indicate predominant processes and also by positioning the dominant lineages to the corresponding arrows.

Response: In response to a suggestion from Reviewer #2, this figure has been removed.

B) Validity:

No flaws were detected.

C) Originality and significance:

The results are interesting, but the conclusions mostly confirm previous studies (Arndt et al., 2013; Beulig et al., 2018). The hydrocarbon degradation element could have been better presented.

Response: The revised manuscript now focusses more explicitly on the role of uncultured microbes, to showcase the novelty of our findings. See above responses to the specific comments.

D) Data & methodology:

The data are satisfactorily detailed and transparent

E) Appropriate use of statistics and treatment of uncertainties:

Comparison between samples could benefit from an associated similarity score

Response: See above response to the specific comments.

F) Conclusions:

Conclusions seem robust enough although they could benefit from better data integration and visualization.

Response: See above response to the specific comments.

G) Suggested improvements:

Suggested improvements are mentioned in the sections above.

H) References:

All references are appropriate

I) Clarity and context:

For the most part it is clear and accessible. The authors could have done a better job at highlighting the novelty and aim of the study and how that differs from previously published studies.

Response: The revised manuscript now focusses on the roles of uncultured microbes. See above responses to the specific comments.

REVIEWERS' COMMENTS:

Reviewer #1 (Remarks to the Author):

The authors have done a very good job in revising and significantly improving the manuscript vs. the original version. I appreciate that changing the title and rationale of the manuscript towards the broader metabolic potential of uncultured bacteria and archaea reflects the findings of the study much better. The decision to conduct a more in-depth metabolomics analysis and to include the new Figure 4 is salutary, and greatly strengthens the metagenome-based predictions. Detection of succinate derivatives indeed offers direct support for the presence of active hydrocarbon-degrading microorganisms. The revised manuscript reads very well, and the major headlines are better connected. In view of these changes, I do support publication of the revised manuscript. A big compliment to the authors for the complexity of the work and thoroughness of the revision. A final remark regarding a minor correction: on line 172 replace 'isopropane' with 'propane': propane has no structural isomers; and on the y axis of Figure 4, replace 'isopropylsuccinate' with '2-isopropylsuccinate', to better reflect the structure of the compound.

Florin Musat
Leipzig, Feb 18, 2019

Reviewer #2 (Remarks to the Author):

Thanks for the good responses. Looks like a nice paper, and I hope it's published.

Reviewer #3 (Remarks to the Author):

The revised manuscript has significantly improved, and many disambiguates from the original version have been resolved. Also, the authors did a good job at satisfactorily addressing the reviewer's comments. Once finalized, this piece of work is going to be a valuable resource for other experimentalists studying microbial metabolic processes in deep water sediments.

Some additional remarks based on the revised manuscript are shown below:

The title suggests that these samples were obtained from hydrocarbon seeps. Was there any direct evidence for hydrocarbon seepage (e.g. was there a microbial mat on top of the sediments, a typical indicator of cold seeps)? If not, could the observed hydrocarbons be the result of a fallout plume as described by Valentine et al 2014 (<https://doi.org/10.1073/pnas.1414873111>)? Unless a more direct evidence for oil seepage is provided, I suggest the title is revised to something less explicit e.g. "Metabolic potential of uncultured bacteria and archaea in deep-sea sediments containing petroleum hydrocarbons".

Sulfate concentrations (Supp Note 2) do not follow the same pattern as the *dsrAB* gene counts (Supp Table 7). Sediment E26 seem to have the lowest metabolic potential for sulfate reduction compared to the other two. Nonetheless sulfate in the E26 has the lowest concentration suggesting higher consumption compared to the other two sediments. Could the authors suggest any potential reasons as to why this might be happening? Also based on that, the statement in L242 doesn't seem entirely accurate.

Having large datasets from different sources can pose a real challenge for the authors to integrate and present them in a comprehensive way. Despite the authors' best efforts their data still seem rather disjointed and therefore harder to collectively interpret. Databases like KEGG and MetaCyc allow simultaneous mapping of genomic/metagenomic, proteomic and metabolomic data against reference

pathway maps. Those databases would be a good starting point for a more comprehensive visualization of multiple datasets.

Response to Referees

Reviewer #1 (Remarks to the Author):

The authors have done a very good job in revising and significantly improving the manuscript vs. the original version. I appreciate that changing the title and rationale of the manuscript towards the broader metabolic potential of uncultured bacteria and archaea reflects the findings of the study much better. The decision to conduct a more in-depth metabolomics analysis and to include the new Figure 4 is salutary, and greatly strengthens the metagenome-based predictions. Detection of succinate derivatives indeed offers direct support for the presence of active hydrocarbon-degrading microorganisms. The revised manuscript reads very well, and the major headlines are better connected. In view of these changes, I do support publication of the revised manuscript. A big compliment to the authors for the complexity of the work and thoroughness of the revision.

A final remark regarding a minor correction: on line 172 replace ‘isopropane’ with ‘propane’: propane has no structural isomers; and on the y axis of Figure 4, replace ‘isopropylsuccinate’ with ‘2-isopropylsuccinate’, to better reflect the structure of the compound.

Response: Thank you very much for your feedback. We now replaced ‘isopropane’ with ‘propane’ in the text and ‘isopropylsuccinate’ with ‘2-isopropylsuccinate’ in Figure 4.

Reviewer #2 (Remarks to the Author):

Thanks for the good responses. Looks like a nice paper, and I hope it's published.

Response: Thank you very much for your feedback.

Reviewer #3 (Remarks to the Author):

The revised manuscript has significantly improved, and many disambiguates from the original version have been resolved. Also, the authors did a good job at satisfactorily addressing the

reviewer's comments. Once finalized, this piece of work is going to be a valuable resource for other experimentalists studying microbial metabolic processes in deep water sediments.

Some additional remarks based on the revised manuscript are shown below:

The title suggests that these samples were obtained from hydrocarbon seeps. Was there any direct evidence for hydrocarbon seepage (e.g. was there a microbial mat on top of the sediments, a typical indicator of cold seeps)? If not, could the observed hydrocarbons be the result of a fallout plume as described by Valentine et al 2014 (<https://doi.org/10.1073/pnas.1414873111>)? Unless a more direct evidence for oil seepage is provided, I suggest the title is revised to something less explicit e.g. "Metabolic potential of uncultured bacteria and archaea in deep-sea sediments containing petroleum hydrocarbons".

Response: The direct evidence is that the hydrocarbons are thermogenic. They are detected at different depths in the bottom half of the piston cores (3-5 mbsf) so they are not from a fallout plume associated with the DWH oil spill. The sampling expedition did not include ROV dives that would be necessary to detect *Beggiatoa* mats as evidence of hydrocarbon seepage. Therefore, the authors think that the current title is a good fit for this study.

*Sulfate concentrations (Supp Note 2) do not follow the same pattern as the *dsrAB* gene counts (Supp Table 7). Sediment E26 seem to have the lowest metabolic potential for sulfate reduction compared to the other two. Nonetheless sulfate in the E26 has the lowest concentration suggesting higher consumption compared to the other two sediments. Could the authors suggest any potential reasons as to why this might be happening? Also based on that, the statement in L242 doesn't seem entirely accurate.*

Response: We agree that sulfate reduction rates might be highest at Site E26, as evidenced from the lowest sulfate concentrations among these three sites. However, sulfate reduction rate measurements (e.g. ³⁵S radiotracer) were not performed at sea. We do not think the lowest *dsrAB* gene read counts necessarily have to correlate with lowest sulfate reduction rate, as we are not inferring activity from metagenomic data. We therefore do not want to link gene level to metabolic activity (i.e. sulfate reduction). We agree with the reviewer's suggestion concerning L242, and have made this statement more accurate by changing "widespread" to "present".

Having large datasets from different sources can pose a real challenge for the authors to integrate and present them in a comprehensive way. Despite the authors' best efforts their data still seem rather disjointed and therefore harder to collectively interpret. Databases like KEGG and MetaCyc allow simultaneous mapping of genomic/metagenomic, proteomic and metabolomic data against reference pathway maps. Those databases would be a good starting point for a more comprehensive visualization of multiple datasets.

Response: It is indeed challenging to integrate all different kinds of omics data into one map. Here we mainly employ metagenomics to predict metabolic potential of uncultured bacteria and archaea. We incorporated metabolomic data to validate specific predictions arising from the metagenomes. Supporting metabolomics are presented close to metagenomic predictions in the main text. We chose not to use databases like KEGG and MetaCyc to reconstruct pathways, as there are too few compounds identified for whole pathway mapping. For example, for alkyl benzene degradation we only identified 7 out of 23 possible metabolites which were too few to create an accurate metabolic pathway. Additionally, for carbon metabolism, from the TCA cycle only fumarate and succinate were identified, and for the beta-oxidation pathway only 10-hydroxydecanoate was identified. Furthermore, certain pathways and metabolisms of interest are not accurately covered in KEGG. For example, hydrogenases are more nuanced and diverse than what KEGG database currently contains.